# PlayerOne: Egocentric World Simulator

**Yuanpeng Tu**[1,2] *    **Hao Luo**[2,3]    **Xi Chen**[1]    **Xiang Bai**[4]    **Fan Wang**[2]    **Hengshuang Zhao**[1,†]

[1]HKU    [2]DAMO Academy, Alibaba Group    [3]Hupan Lab    [4]HUST

*https://playerone.github.io*

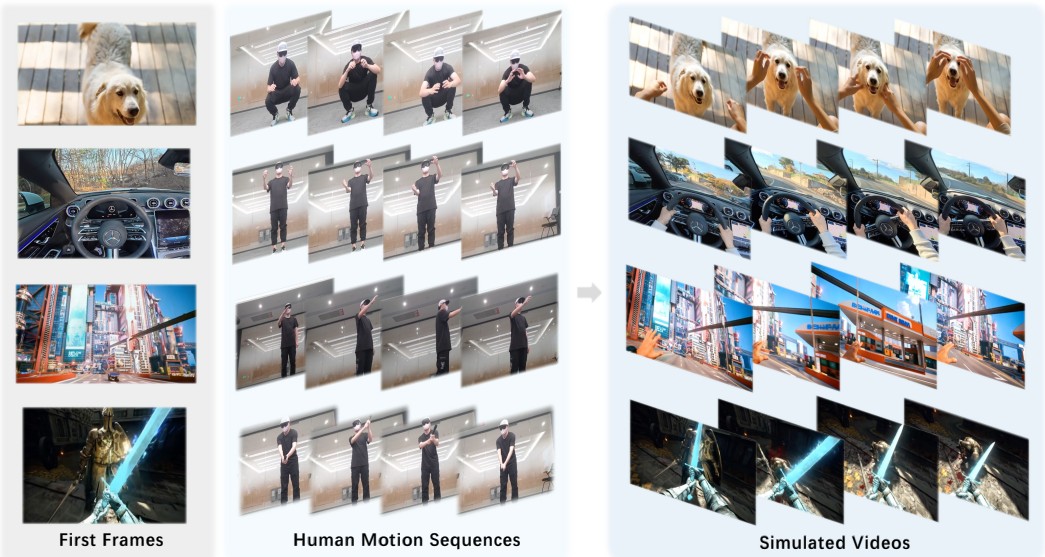

Figure 1: **Simulated videos of our PlayerOne**. Given an egocentric image as the scene to be explored, we can simulate egocentric immersive videos that are accurately aligned with the user's motion sequence captured by an exocentric camera. All the users have been anonymized and action videos are shot with the front camera.

## Abstract

We introduce PlayerOne, the first egocentric realistic world simulator, facilitating immersive and unrestricted exploration within vividly dynamic environments. Given an egocentric scene image from the user, PlayerOne can accurately construct the corresponding world and generate egocentric videos that are strictly aligned with the real-scene human motion of the user captured by an exocentric camera. PlayerOne is trained in a coarse-to-fine pipeline that first performs pretraining on large-scale egocentric text-video pairs for coarse-level egocentric understanding, followed by finetuning on synchronous motion-video data extracted from egocentric-exocentric video datasets with our automatic construction pipeline. Besides, considering the varying importance of different components, we design a part-disentangled motion injection scheme, enabling precise control of part-level movements. In addition, we devise a joint reconstruction framework that progressively models both the 4D scene and video frames, ensuring scene consistency in the long-form video generation. Experimental results demonstrate its great generalization ability in precise control of varying human movements and world-consistent modeling of diverse scenarios. It marks the first endeavor into egocentric real-world simulation and can pave the way for the community to delve into fresh frontiers of world modeling and its diverse applications.

---

*Work during DAMO Academy internship. † Corresponding author.

39th Conference on Neural Information Processing Systems (NeurIPS 2025).

# 1  Introduction

World models [36, 28, 35, 13, 1, 34] have undergone extensive research due to their ability to model environmental dynamics and predict long-term outcomes. Recent breakthroughs in video diffusion models [30, 12, 17] have revolutionized this domain, enabling the synthesis of high-fidelity, action-conditioned simulations that forecast intricate future states. These advancements empower applications ranging from autonomous navigation in dynamic real-world environments to the creation of immersive, responsive virtual worlds in AAA game development. By bridging the gap between predictive modeling and interactive realism, world simulators are emerging as critical infrastructure for next-generation autonomous systems and game engines, particularly in scenarios requiring real-time adaptation to complex, evolving interactions.

Despite significant progress, this topic remains underexplored in existing research. Prior studies [9, 36, 7] predominantly focused on simulations within game-like environments, falling short of replicating realistic scenarios. Additionally, in their simulated environments, users are limited to performing predetermined actions (*i.e.*, directional movements). Operating within the confines of a constructed world restricts the execution of unrestricted movements as in real-world scenarios. While some initial efforts [19, 26, 1] have been made toward real-world simulation, they mainly contribute to world-consistent generation without human movement control. Consequently, users are reduced to passive spectators within the environment, rather than being active participants. This limitation significantly impacts the user experience, as it prevents the establishment of a genuine connection between the user and the simulated environment.

Faced with these challenges, we aim to design an egocentric world foundational framework that enables the user being a freeform adventurer. Given a user-provided egocentric image as the world to be explored, it can enable the user to perform unrestricted human movements real-time captured by an exocentric camera and consistent 4D scene modeling in the simulated world. Specifically, we propose the first realistic egocentric world simulator termed PlayerOne. Starting from a diffusion transformer (DiT) model [23], we first extract the latent of an egocentric user-input image. Meanwhile, we select the real-world human motions (*i.e.*, human pose or keypoints) as our motion representation. Considering the varying importance of different body parts in our task, the human motion sequence is partitioned into three groups (*i.e.*, head, hands, feet and body) and fed into our part-disentangled motion injection to generate latents that can enable precise part-wise control. Additionally, we developed a joint scene-frame reconstruction framework that can progressively complete scene point maps during the video generation process to enable scene-consistent generation. The DiT model takes the concatenation of the first frame latent, motion latent, video latent, and the point map latent as input and conducts noising and denoising on both the video and point map latent. Notably, the point map sequence is not required during inference, ensuring practical efficiency. Moreover, to overcome the absence of publicly available datasets, we curate required motion-video pairs from existing egocentric-exocentric datasets using an automated pipeline designed to filter and retain high-quality data. A coarse-to-fine training strategy is also designed to compensate for the data scarcity. The base model is fine-tuned on large-scale egocentric text-video data for coarse-level generation, then refined on our curated dataset to achieve precise motion control and scene modeling. Finally, we distill our trained model [38] to achieve real-time generation. By integrating these innovations, PlayerOne advances the field of dynamic world modeling. Our contributions are summarized as follows:

- We introduce PlayerOne, the first egocentric foundational simulator for realistic worlds, capable of generating video streams with precise control of highly free human motions and world consistency in real-time and exhibiting strong generalization in diverse scenarios.
- We design a novel part-disentangled motion injection scheme to enhance fine-grained motion alignment, where a joint scene-frame reconstruction framework is introduced to guarantee world-consistent modeling in long-term video generation as well.
- We construct an effective automatic dataset construction pipeline to extract high-quality motion-video pairs from existing egocentric-exocentric datasets, where a coarse-to-fine training scheme is also introduced to compensate for the data scarcity.

# 2  Related Work

**Video generation.** The rapid development of diffusion models [27, 14, 41, 3] has driven substantial advancements in video generation. Early researchers [10, 5] adapted existing text-to-image models to

enable text-to-video generation to compensate for the limited availability of high-quality video-text datasets. Subsequently, diffusion transformers based frameworks [37, 23, 17, 12, 30, 29] are proposed. When scaling-up training, they enable more highly realistic and temporally coherent generation results. Among them, HunyuanVideo [17] substitutes T5 with a Multimodal Large Language Model. LTX-Video [12] modifies the VAE decoder to handle the final denoising step and convert latents into pixels. Wan [30] introduces a full spatial-temporal attention to ensure computational efficiency.

**World models.** Existing world models [13, 35, 36, 7, 1, 34] can be roughly divided into two categories: 1) Agent learning targeted models, 2) World simulation models. For the former, they [13, 35, 34] aim at enhancing policy learning within simulated environments. Among them, Dreamer [13] and DayDreamer [35] solve long-horizon tasks from images purely by latent imagination. MuZero [34] runs the self-play of Monte Carlo tree search to build world models for Atari. Distinct from this direction, world simulation aims to model an environment by predicting the next state given the current state and action. These works focus on human interaction with neural networks through high-quality rendering, robust control, and strong domain generalization to real-world scenarios. With advances in video generation, high-quality world simulation with robust control has become feasible, leading to numerous works focusing on interactive world simulation [9, 26, 19, 36, 7, 1, 28, 6]. Among these works, WORLDMEM [36]. The Matrix [7] proposes the first world simulator capable of generating infinitely long real-scene video streams with real-time, responsive control. Matrix-Game [43] redefines video generation as an interactive process of exploration and creation. Cosmos [1] presents a general-purpose world model and a pre-training-then-post-training scheme. Aether [28] designs a unified framework with synergistic knowledge sharing across reconstruction, prediction, and planning objectives. However, these methods primarily focus on virtual game scenarios and are limited to specific directional actions, rather than facilitating high-degree-of-freedom motion control in real-world environments. To address these limitations, we target at developing a human motion driven realistic world simulator. Given an egocentric image, we can construct a real-scene world that immerses users as freeform adventurers with precise and unrestricted human motion control.

## 3 Method

In this section, we detail the methodology of PlayerOne. Sec. 3.1 introduces the relevant preliminaries and the overall pipeline. Sec. 3.2 presents the core of our proposed model, followed by Sec. 3.3 introducing our dataset construction and training strategy.

### 3.1 Overview

Video diffusion models [23, 15] consist of two key processes: a forward (noising) process and a reverse (denoising) process. The forward process gradually adds Gaussian noise, denoted as $\epsilon \sim \mathcal{N}(0, \mathbf{I})$, to a clean latent sample $z_0 \in \mathbb{R}^{k \times c \times h \times w}$, where k, c, h, and w represent the dimensions of the video latents. This transforms $z_0$ into a noisy latent $z_t$. In the reverse process, a learned denoising model $\epsilon_\theta$ progressively removes the noise from $z_t$ to reconstruct the original latent representation.

As shown in Fig. 2, our method comprises two core modules: Part-disentangled Motion Injection (PMI) and Scene-frame Reconstruction (SR). In PMI, we use real-scene human motion as the motion condition to enable free action control for the user. The first frame is converted into $z_{frame}$ via a 3D VAE encoder. The human motion sequence is split into three parts based on varying importance, and each part is fed into a 3D motion encoder to obtain latents. These motion latents are concatenated into $z_{motion} \in \mathcal{R}^{k \times 3 \times h \times w}$. To improve view alignment, we transform the head parameters of the human motion sequence into a camera sequence, which is then fed into a camera encoder. The output is added to the noised video latents $z_{video}$ to inject view-change signals. In SR, we jointly reconstruct video frames and 4D scenes to ensure world-consistent generation in the context of long video generation. We render a point map sequence from the ground truth video and feed it into a point map encoder with an adapter to obtain $z_{point} \in \mathcal{R}^{k \times 64 \times h \times w}$. Finally, all latents and conditions are concatenated channel-wise. The training objective of our method can be expressed as:

$$\mathcal{L} = \mathbb{E}_{t \sim \mathcal{U}(0,1), \epsilon \sim \mathcal{N}(0,\mathbf{I})} \left[ \| \epsilon - \epsilon_\theta \left( \mathbf{z}_t, t \right) \|_2^2 \right], \quad \text{where} \quad z_t = \alpha_t z_0 + \delta_t \epsilon \quad (1)$$

Where $t = 1, \ldots, T$, $\alpha_t{}^2 + \delta_t{}^2 = 1$. Since we only add noise to point map latents and video latents, thus $z_0 = z_{video} \otimes z_{point}$. $\otimes$ denotes the channel-wise concatenation operation, $\mathcal{U}(\cdot)$ represents a uniform distribution, and T denotes the denoising steps.

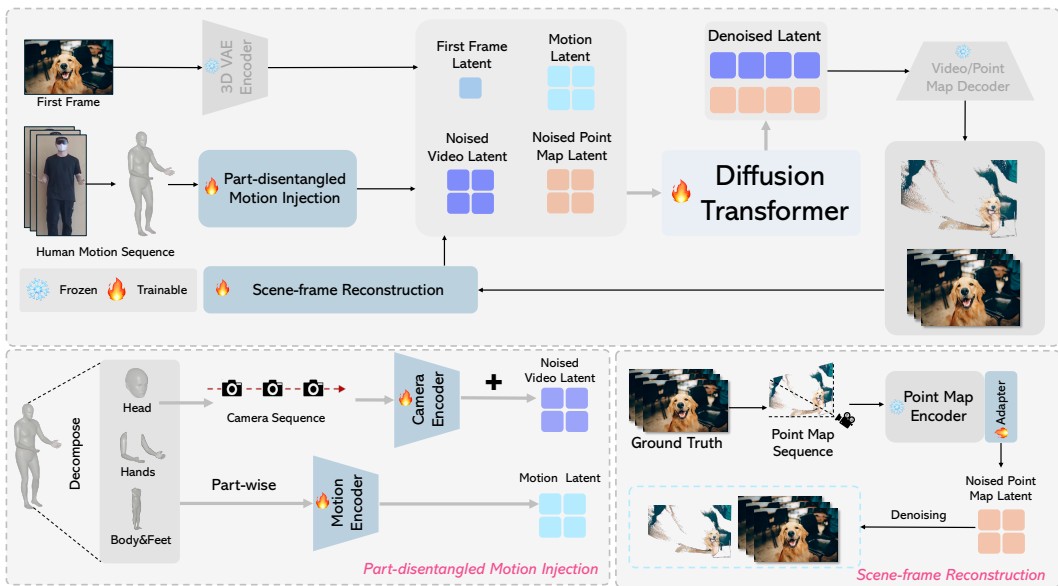

Figure 2: **Overall framework of our PlayerOne**. It begins by converting the egocentric first frame into visual tokens. The human motion sequence is split into groups and fed into the motion encoders respectively to generate part-wise motion latents, with the head parameters converted into a rotation-only camera sequence. This camera sequence is then encoded via a camera encoder, and its output is injected into noised video latents to improve view-change alignment. Next, we render a 4D scene point map sequence with the ground truth video, which is then processed by a point map encoder with an adapter to produce scene latents. Then we input the concatenation of these latents into the DiT Model and perform noising and denoising on both the video and scene latents to ensure world-consistent generation. Finally, the denoised latents are decoded by VAE decoders to produce the final results. Note that only the first frame and the human motion sequence are needed for inference.

## 3.2 Model Components

**Part-disentangled motion injection.** Prior studies [19, 26, 7, 43] typically utilize camera trajectories as motion conditions or are constrained to specific directional movements. These restrictions confine users to passive "observer" roles, preventing meaningful user interaction. In contrast, our approach empowers users to become active "participants" by adopting real-world human motion sequences (*i.e.*, human pose or keypoints) as motion conditions, allowing for more natural and unrestricted movement. However, our empirical analysis reveals that extracting latent representations holistically from human motion parameters complicates precise motion alignment. To address this challenge, we introduce a part-disentangled motion injection strategy that recognizes the distinct roles of various body parts. Specifically, hand movements are essential for interacting with objects in the environment, while the head plays a crucial role in maintaining egocentric perspective alignment. Accordingly, we categorize the human motion parameters into three groups: body and feet, hands, and head. Each group is processed through its own dedicated motion encoder, comprising eight layers of 3D convolutional networks, to extract the relevant latent features. This specialized processing ensures accurate and synchronized motion alignment. These latents are subsequently concatenated along the channel dimension to form the final part-aware motion latent representation $z_{motion} \in \mathcal{R}^{k \times 3 \times h \times w}$.

To further enhance the egocentric view alignment, we solely transform the head parameters of the human motion sequence into a sequence of camera extrinsics with only rotation values. We zero out the translation values in the camera extrinsics, assuming the head parameters are at the camera coordinate system's origin. Specifically, suppose the head parameter $\mathbf{v} = (\theta_x, \theta_y, \theta_z)$, we first normalize the rotation axis as follows:

$$\mathbf{u} = \frac{\mathbf{v}}{\|\mathbf{v}\|}, \quad \theta = \|\mathbf{v}\| \tag{2}$$

Then we construct the rotation matrix as follows:

$$\mathbf{R} = \mathbf{I} + \sin\theta \cdot [\mathbf{u}]_\times + (1 - \cos\theta) \cdot [\mathbf{u}]_\times^2 \tag{3}$$

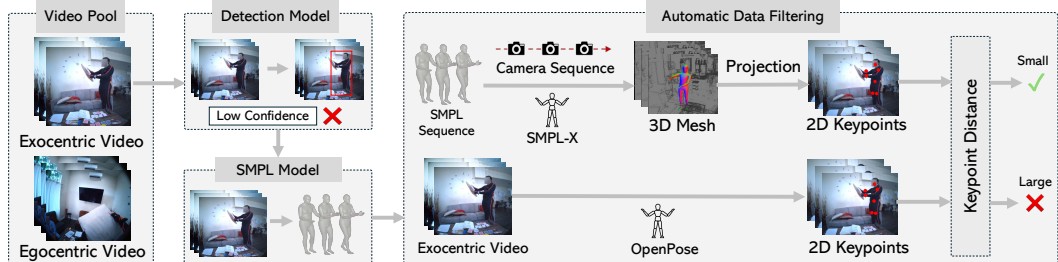

Figure 3: **The overall pipeline of the dataset construction**. By seamlessly integrating detection and human pose estimation models, we can extract motion-video pairs from existing egocentric-exocentric video datasets while retaining high-quality data through our automatic filtering scheme.

Where $\mathbf{u}_\times$ is the cross product matrix of $\mathbf{u}$, which can be denoted as follows:

$$[\mathbf{u}]_\times = \begin{bmatrix} 0 & -u_z & u_y \\ u_z & 0 & -u_x \\ -u_y & u_x & 0 \end{bmatrix} \tag{4}$$

Then we use Plücker ray [40] to parameterize the camera extrinsics and then feed the output to an extra camera encoder, which shares a similar structure with the motion encoder. Then the latents from this encoder are added to the noised video latents to inject the view-change information.

**Scene-frame reconstruction.** While PMI enables precise control over egocentric perspective and motion, it does not guarantee scene consistency within the generated world. To address this limitation, we introduce a joint reconstruction framework that simultaneously models the 4D scene and video frames, ensuring scene coherence and continuity throughout the video. Specifically, it begins by employing CUT3R [31] to generate a point map for each frame based on ground truth video data, reconstructing the n-th frame's point map using information from frames 1 through n. These point maps are then compressed into latent representations using a specialized point map encoder [16]. To integrate these latents with video features, we implement an adapter composed of five 3D convolutional layers. This adapter aligns the point map latents with video latents and projects them into a shared latent space, facilitating seamless integration of motion and environmental data. Finally, we concatenate the latent representations from the first frame, the human motion sequence, the noised video latents, and corresponding noised point map latents. This comprehensive input is then fed into a diffusion transformer for denoising, resulting in a coherent and visually consistent world. Importantly, point maps are only required during the training phase. *During inference, the system simplifies the process by utilizing only the first frame and the corresponding human motion sequence to generate world-consistent videos.* This streamlined approach enhances generation efficiency while ensuring that the resulting environment remains stable and realistic throughout the entire video.

### 3.3 Training Strategy

**Dataset preparation.** The ideal training samples for our task are egocentric videos paired with corresponding motion sequences. However, no such dataset currently exists in publicly available repositories. As a substitute, we derive these data pairs from existing egocentric-exocentric video datasets through an automatic pipeline. Specifically, for each synchronized egocentric-exocentric video pair, we first employ SAM2 [25] to detect the largest person in the exocentric view. The background-removed exocentric video is then processed using SMPLest-X [39] to extract the SMPL parameters of the identified individual as the human motion. To enhance optimization stability, an L2 regularization prior is incorporated. We then evaluate the 2D reprojection consistency to filter out low-quality SMPL data. This involves generating a 3D mesh from the SMPL parameters using SMPLX [22], projecting the 3D joints onto the 2D image plane with the corresponding camera parameters, and extracting 2D key points via OpenPose [4]. The reprojection error is calculated by measuring the distance between the SMPL-projected 2D key points and those detected by OpenPose. Data pairs with reprojection errors in the top 10% are excluded, ensuring a final dataset of high-quality motion-video pairs. The refined SMPL parameters are decomposed into body and feet (66 dimensions), head orientation (3 dimensions), and hand articulation (45 dimensions per hand) components for each frame. These components are fed into their respective motion encoders. The dataset construction pipeline is illustrated in Fig. 3. As detailed in Tab. 1, our training dataset combines multiple publicly available datasets to ensure comprehensive coverage of diverse environmental contexts, action types, and intensity levels, thereby enhancing model generalization.

Table 1: **Statistics of datasets** used for training our PlayerOne. "quality" particularly refer to the image resolution. "Ego-Exo" denotes whether the dataset contains egocentric-exocentric video pairs.

| Dataset | EgoExo-4D [8] | Nymeria [20] | FT-HID [11] | EgoExo-Fitness [18] | Egovid-5M [33] |
|---|---|---|---|---|---|
| Size | 740 | 1,200 | 38,364 | 1,276 | 5M |
| Resolution | 1080p | 1408p | 1080p | 1080p | 1080p |
| Ego-Exo | ✓ | ✓ | ✓ | ✓ | × |

Figure 4: **Investigation on coarse-to-fine training**. "Joint-Train" and "No Pretrain" denote training with both motion-video pairs and large-scale egocentric videos in a one-stage manner and training with only motion-video pairs respectively. The Wanx2.1 1.3B is adopted as the baseline.

**Coarse-to-fine training.** Though we can extract high-quality motion-video training data with our automatic pipeline, the limited scale of this dataset is insufficient for training video generation models to produce high-quality egocentric videos. To address this, we harness the extensive egocentric text-video datasets (*i.e.*, Egovid-5M [33]). Specifically, we first fine-tune the baseline model using LoRA on large-scale egocentric text-video data pairs, enabling egocentric video generation with coarse-level motion alignment. Then we freeze the trained LoRA and fine-tune the last six blocks of the model with our constructed high-quality dataset to enhance fine-grained human motion alignment and view-invariant scene modeling, which can effectively address the scarcity of pair-wise data. Finally, we adopt an asymmetric distillation strategy that supervises a causal student model with a bidirectional teacher [38] to achieve real-time generation and long-duration video synthesis.

## 4 Experiments

### 4.1 Experimental Setting

**Implementation details.** We choose Wanx2.1 1.3B [30] as the base generator. We set the LoRA rank and the update weight of the matrices as 128 and 4 respectively and initialize its weight following [30]. The inference step and the learning rate are set as 50 and $1 \times 10^{-5}$ respectively, where the Adam optimizer and mixed-precision bf16 are adopted. The cfg of 7.5 is used. We train our model for 100,000 steps on 8 NVIDIA A100 GPUs with a batch size of 56 and sample resolution of $480 \times 480$. The generated video runs at eight frames per second, and we utilize 49 video frames (6 seconds) for training. After distillation, our method can achieve 8 FPS to generate the desired results. *All the action videos in this paper are shot with the front camera.*

**Benchmark.** Since there is no publicly available benchmark for our task, we construct a benchmark with 100 videos collected from Nymeria [21] dataset, which is not included for training. It consists of coarse-level motion descriptions for each sample and covers diverse realistic scenarios. Considering the information gap between the human motion sequence and the text, we further use Qwen2.5-VL [2] to enrich the caption to generate videos for the competitors for more fair comparisons.

**Metrics.** On our constructed benchmark, for evaluation of alignment with the given text descriptions, we calculate both CLIP-Score and DINO-Score, where LPIPS [42] is employed to evaluate the video fidelity of the generated video. Besides, we calculate the frame consistency to evaluate the temporal coherence and consistency of the generated video frames over time. We further utilize a 3D hand pose estimation model [24] to estimate the hand pose of the generated videos and use the results of

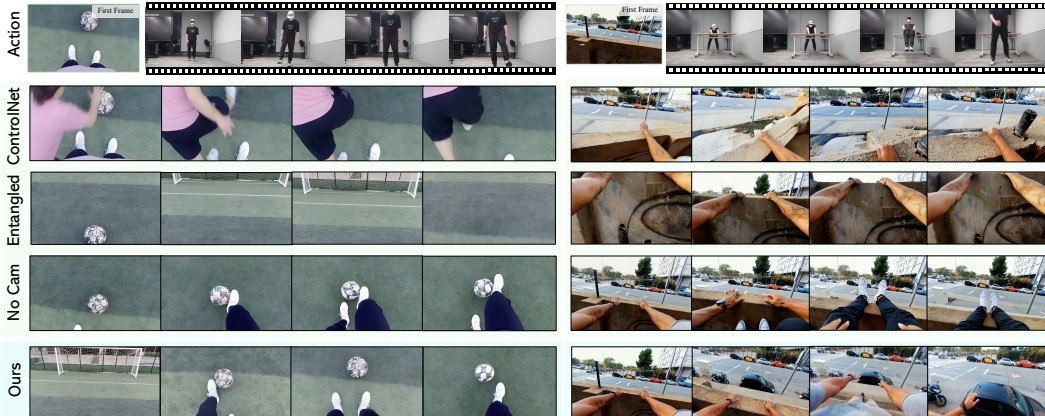

Figure 5: **Investigation on part-disentangled motion injection**. "ControlNet" denotes injecting motion latents with a ControlNet [41]. "Entangled" and "No Cam" denote inputting the whole motion sequence into a motion encoder without dividing into groups and removing the camera encoder respectively.

Table 2: **Quantitative evaluation** on the components of PlayerOne. PlayerOne outperforms all these variants. "No Camera"/"Filtering" denote training without/with the camera encoder/data filtering.

| | DINO-Score (↑) | CLIP-Score (↑) | MPJPE (↓) | MRRPE (↓) | FVD (↓) | LPIPS(↓) |
|---|---|---|---|---|---|---|
| Baseline | 51.3 | 65.6 | 376.14 | 341.01 | 394.16 | 0.1421 |
| + Pretrain | 56.6 | 74.4 | 258.05 | 232.17 | 301.32 | 0.1146 |
| + Pretrain&ControlNet | 57.1 | 75.2 | 241.73 | 218.46 | 287.52 | 0.1103 |
| + Pretrain&Entangled | 58.0 | 76.3 | 235.12 | 212.53 | 279.41 | 0.1060 |
| + Pretrain&PMI (No Camera) | 60.7 | 79.8 | 183.25 | 196.35 | 257.04 | 0.0902 |
| + Pretrain&PMI | 62.5 | 81.3 | 156.76 | 175.18 | 245.72 | 0.0839 |
| + Pretrain&PMI&Filtering | 64.2 | 83.8 | 141.56 | 163.04 | 230.50 | 0.0782 |
| + Pretrain&PMI&Filtering&Recon(No Adapter) | 62.7 | 81.6 | 176.23 | 180.10 | 240.17 | 0.0919 |
| + Pretrain&PMI&Filtering&Recon(DUSt3R) | 67.5 | 87.7 | 129.08 | 152.22 | 228.20 | 0.0685 |
| PlayerOne(ours) | **67.8** | **88.2** | **127.16** | **151.62** | **226.12** | **0.0663** |

the ground truth video as the labels. Afterward, we follow [24] to calculate two metrics: (1) Mean Per-Joint Position Error (MPJPE): the L2 distance between the predicted and ground truth joints for each hand after subtracting the root joint. (2) Mean Relative-Root Position Error (MRRPE): the metric distance between the root joints of the left hand and right hand.

### 4.2 Ablation Study

**Investigation on coarse-to-fine training.** We first evaluate several variants of our coarse-to-fine training scheme, as depicted in Fig. 4. Specifically, when inputting action descriptions into the baseline model without fine-tuning, the generated results exhibit noticeable flaws, such as hand distortions or the unexpected appearance of individuals. Similar issues can be observed when training with only motion-video pairs. We also explore jointly training with both large-scale egocentric videos and motion-video pairs. Specifically, when inputting egocentric videos, we set the motion latent values to zero and extract the latents of the text description to serve as the motion condition, where a balanced-sampling strategy is used as well. Despite this variant being capable of generating egocentric videos, it fails to produce results accurately aligned with the given human motion conditions. In contrast, our coarse-to-fine training scheme delivers much better outcomes compared to these variants.

**Investigation on part-disentangled motion injection.** Next, we conduct a detailed analysis of our PMI module. Specifically, three variants are included: ControlNet-based [41] motion injection, inputting motion sequences as a unified entity (the "Entangled" scheme), and removing our camera encoder. As shown in Fig. 5, the ControlNet-based scheme suffers from information loss, preventing it from producing results that accurately align with the specified motion conditions. Similarly, the entangled scheme demonstrates comparable shortcomings. Furthermore, removing the camera encoder leads to the model's inability to generate view-accurate alignments. As depicted in Fig. 5, this variant fails to produce the corresponding perspective change associated with crouching. Ultimately, our PMI module successfully generates outcomes that are both view-aligned and action-aligned.

**Investigation on scene-frame reconstruction.** Additionally, we conducted a detailed analysis of the SR module, exploring three variants: omitting reconstruction, removing the adapter within the SR

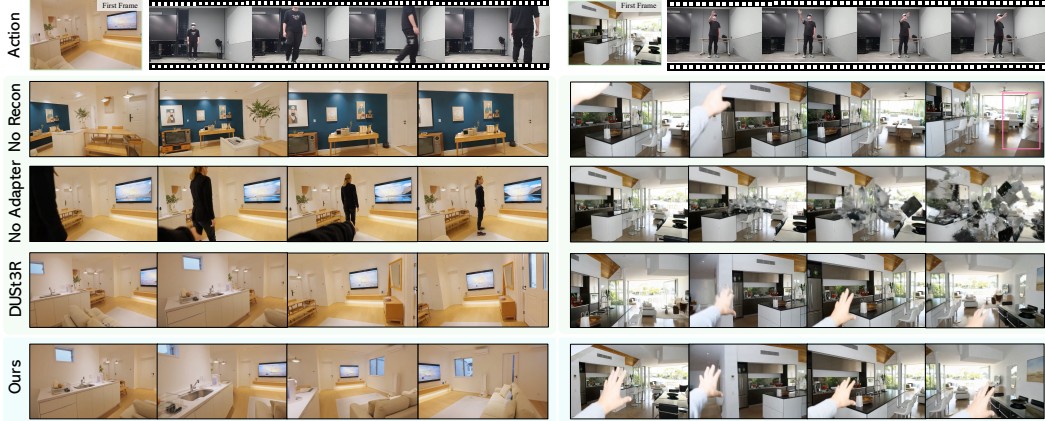

Figure 6: **Investigation on scene-frame reconstruction**. "No Recon"/"No Adapter" denote training without reconstruction/the adapter. "DUStR" is replacing CUT3R with DUStR for point map rendering.

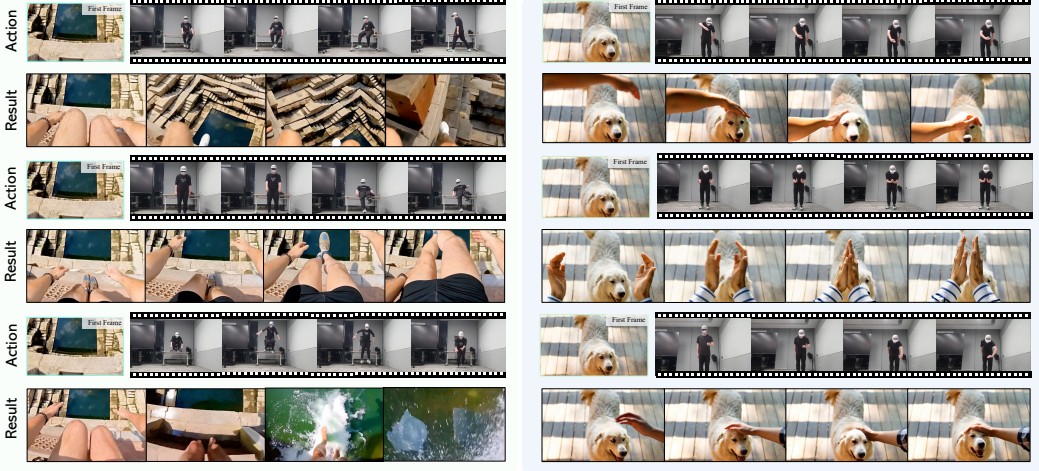

Figure 7: **Qualitative evaluation on the motion alignment**. We generate simulated videos based on the same first frame but different motion sequences. Results show that we can achieve accurate motion alignment.

module, and substituting CUT3R [31] with DUStR [32] for point map rendering. As illustrated in Fig. 6, the absence of reconstruction results in the model's inability to generate consistently simulated results. Moreover, due to the distribution gap between latents of frames and point maps, training without the adapter leads to difficulty in loss convergence, causing noticeable distortions. Furthermore, after replacing CUT3R [31] with DUStR [32], our PlayerOne can also produce scene-consistent outputs, demonstrating its robustness to different point map rendering techniques.

**Motion alignment.** To verify the alignment capability with the given motion condition, we conduct experiments by generating world-simulated videos with the same first frame but different human motion sequences. Fig. 7 shows that our PlayerOne can accurately generate corresponding results according to different conditions and produce reasonable interactive changes.

**Quantitative comparisons.** We provide quantitative results on the core components of our PlayerOne in Tab. 2. All numerical results concur with the visualization outcomes. A significant performance improvement is observed when the model undergoes pre-training on large-scale egocentric text-video datasets. The introduction of PMI yields an additional accuracy boost, and it outperforms all of its variants. In addition, our designed filtering strategy maximizes performance as well by filtering noisy motion-video pairs. After removing the adapter, the performance suffers from a notable degradation due to the distribution gap between latents of video frames and point maps. By introducing our joint scene-frame reconstruction scheme, we achieve superior results across all metrics.

## 4.3 Comparison with State-of-the-arts

**Quantitative comparison.** Since there is no method sharing the same setting as ours, we selected two potential competitors for comparison: Cosmos [1] and Aether [28]. As shown in Tab. 3, our

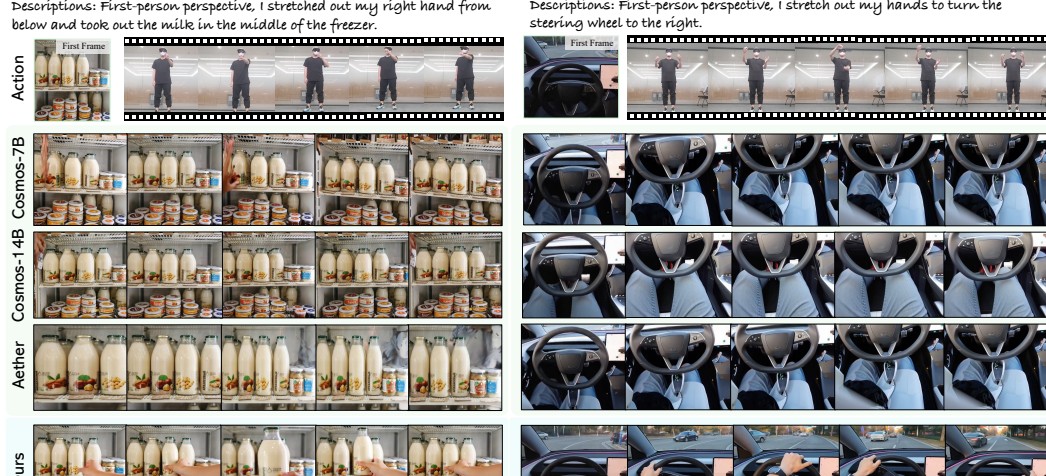

Figure 8: **Qualitative comparisons between our method and other competitors**. Our PlayerOne can achieve the best performance on both the motion alignment, video quality.

Table 3: **Quantitative comparison** between our PlayerOne and other works. Seven metrics are employed for the evaluation. PlayerOne outperforms these methods across all the metrics.

| | DINO-Score (↑) | CLIP-Score (↑) | MPJPE (↓) | MRRPE (↓) | FVD (↓) | LPIPS(↓) |
|---|---|---|---|---|---|---|
| Aether [28] | 38.0 | 64.2 | 415.70 | 431.05 | 397.40 | 0.1856 |
| Cosmos(Diff-7B) [1] | 45.3 | 70.3 | 301.92 | 324.12 | 346.09 | 0.1630 |
| Cosmos(Diff-14B) [1] | 51.6 | 79.7 | 256.73 | 253.06 | 302.17 | 0.1351 |
| PlayerOne(ours) | **67.8** | **88.2** | **127.16** | **151.62** | **226.12** | **0.0663** |

Table 4: **User study** on our PlayerOne and existing alternatives. "Quality", "Fidelity", "Smooth", and "Alignment" measure synthesis quality, object identity preservation, motion consistency, and alignment with the text descriptions, respectively. Each metric is rated from 1 (worst) to 4 (best).

| | Quality (↑) | Fidelity (↑) | Smooth (↑) | Alignment (↑) |
|---|---|---|---|---|
| Aether [28] | 1.32 | 1.30 | 1.31 | 1.34 |
| Cosmos(Diff-7B) [1] | 2.07 | 2.13 | 2.05 | 2.09 |
| Cosmos(Diff-14B) [1] | 3.02 | 2.94 | 2.98 | 2.71 |
| PlayerOne(ours) | **3.59** | **3.63** | **3.65** | **3.86** |

PlayerOne outperforms all the baselines by large margins, especially on the metrics of motion alignment. Notably, Cosmos [1] exhibits better generalization ability than Aether [28] by explicitly capturing general knowledge of real-world physics and natural behaviors. Besides qualitative results, we provide visualization comparisons in Fig. 8, where consistent superiority can be observed in diverse scenarios for both user interaction and world modeling.

**User study.** In Tab. 4, we report the comparison results of human preference rates. We let 20 annotators rate 25 groups of videos, where each group contains the generated video of each method and text description. And we provide detailed regulations to rate the results for scores of 1-4 from four views: "Quality", "Smooth", "Fidelity", "Alignment". "Quality" counts for whether the result is harmonized without considering fidelity. "Smooth" assesses the motion consistency across the video. "Fidelity" measures ID preservation and distortions within the video, while we use "Alignment" to measure the alignment with the given text descriptions. It can be noted that our model demonstrates significant superiority across all the metrics, especially for "Alignment", and "Smooth".

## 5 Conclusion

In conclusion, PlayerOne represents a significant advancement in interactive and realistic world modeling for video generation. Unlike conventional models that are restricted to particular game scenarios or actions, our PlayerOne can capture the complex dynamics of general-world environments and enable free motion control within the simulated world. By formulating world modeling as a joint process of videos and 4D scenes, our PlayerOne ensures coherent world generation and enhances motion and view alignment with the given conditions through part-disentangled motion injection. Experimental results demonstrate our superior performance across diverse scenarios.

**Limitations.** Despite the compelling outcomes, our performance in game scenarios is slightly inferior to realistic ones, likely due to the imbalanced distribution between realistic and game training data. It can be addressed by incorporating more game-scenario datasets in future research.

**Acknowledgements.**   This work was supported by the National Natural Science Foundation of China (No. 62441615, 62422606, 62201484) and DAMO Academy via DAMO Academy Research Intern Program.

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

Figure 9: **The reconstructed scene of our PlayerOne**. Our PlayerOnecan achieve relatively precise scene reconstruction by jointly modeling of video frames and scenes.

# Appendix

## Appendix Overview

This appendix complements the main paper with additional results, analyses, and implementation details:

- **Visualizations and reconstructed scenes.** Qualitative roll-outs and scene reconstructions in diverse scenarios (Figs. 9, 10, 11–13, 14).
- **Implementation notes.** Clarifications of the training/inference pipeline (world/scene consistency is preserved at inference), superiority over prior video diffusion models, and detailed distillation settings.

## A   More Experimental Results

**Framework comparisons with prior works.**   Fig. 10 illustrates the framework comparison between our PlayerOne and other competitors. Prior studies [9, 36, 7] have mainly concentrated on simulations within game-like environments, yet they often fail to accurately replicate real-world scenarios. Within these simulated environments, users are typically restricted to performing predefined actions, such

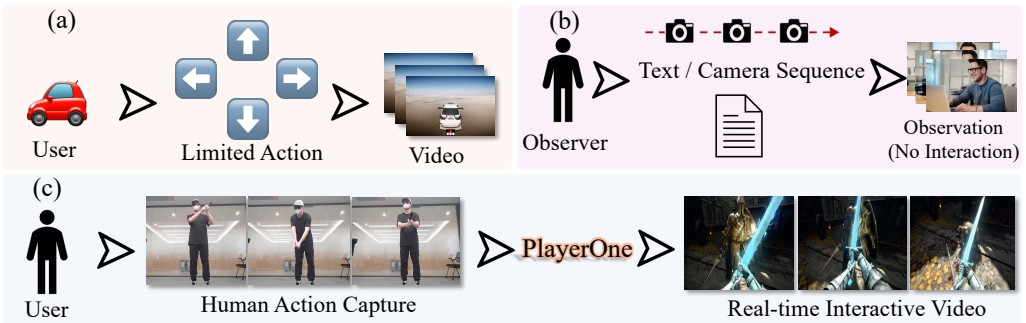

Figure 10: **Difference between our PlayerOneand prior works**. Our PlayerOnecan enable freeform movements in the simulated world and achieve great world consistency across diverse scenarios.

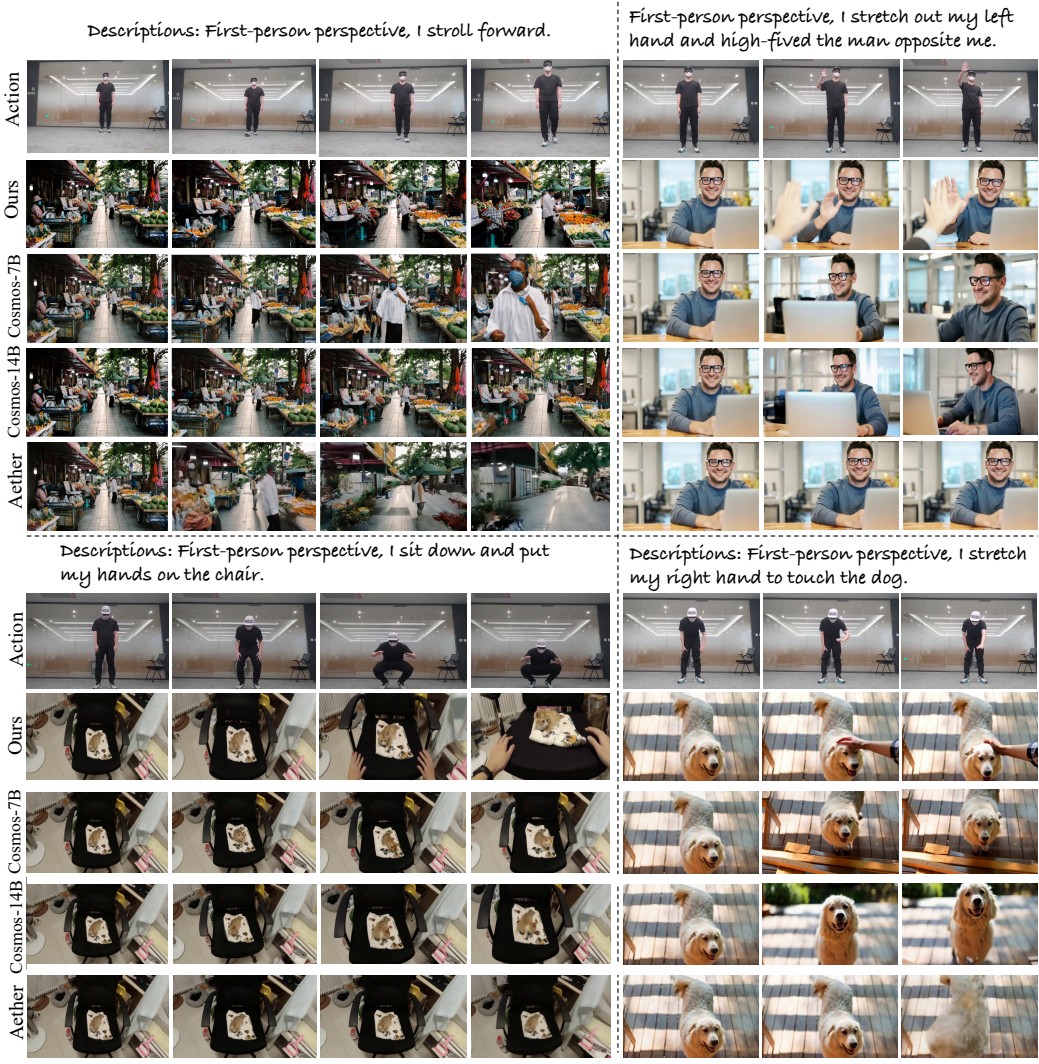

Figure 11: **More visualization results generated by our PlayerOne**. Our method demonstrates great superiority in both motion alignment and environmental interaction across different domains.

as directional movements. This limitation confines user interactions to a constructed world, thereby

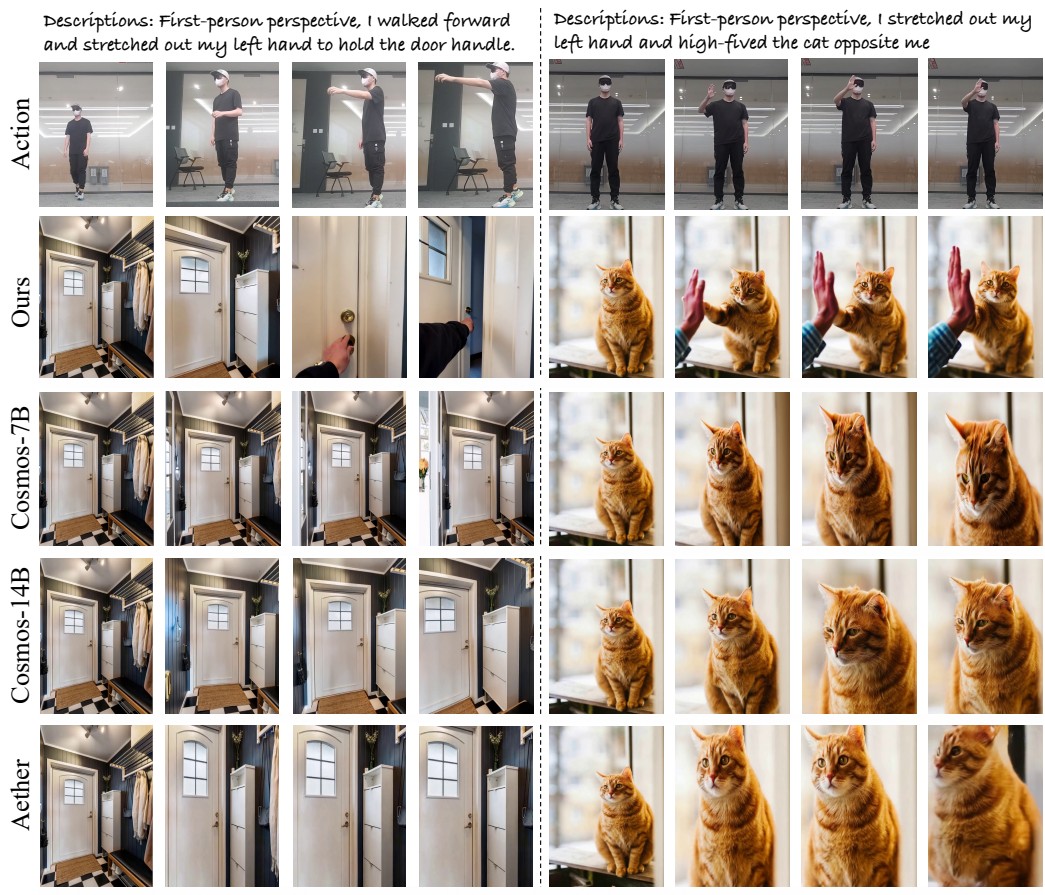

Figure 12: **More visualization results generated by our PlayerOne**. Our method demonstrates great superiority in both motion alignment and environmental interaction across different domains.

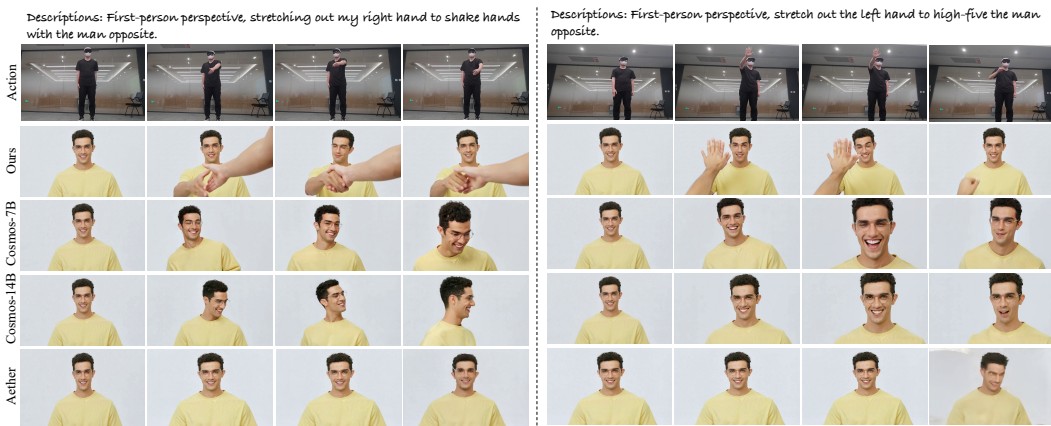

Figure 13: **More visualization results generated by our PlayerOne**. Our method demonstrates great superiority in both motion alignment and environmental interaction across different domains.

restricting the execution of freeform movements akin to those in real-world settings. Existing realistic world simulators [19, 26, 1] often focus solely on world-consistent generation, lacking mechanisms for human movement control. As a result, users are relegated to passive observers, rather than active participants, within the environment. This significantly impacts the user experience by hindering the

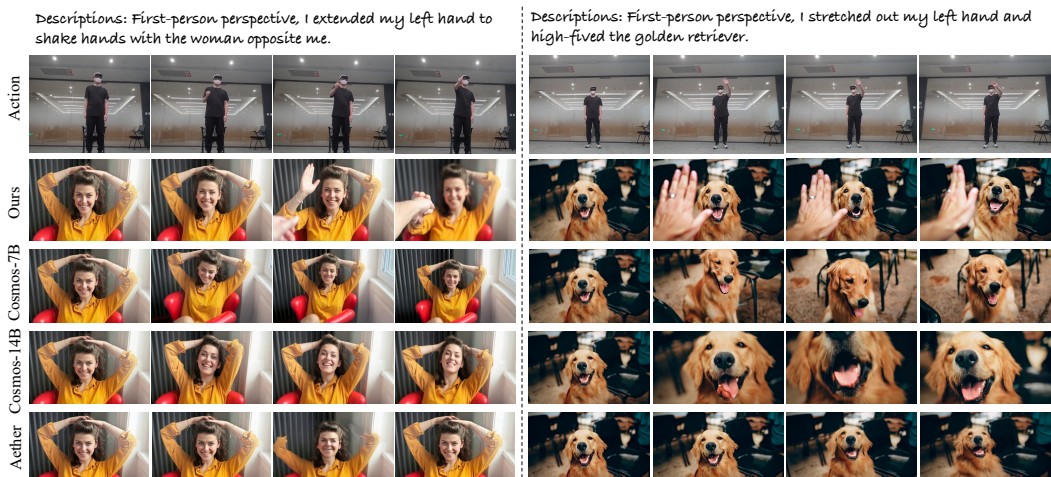

Figure 14: **More visualization results generated by our PlayerOne**. Our method demonstrates great superiority in both motion alignment and environmental interaction across different domains.

Table 5: **Quantitative evaluation** on the components of PlayerOne on the Nymeria dataset. "No Camera"/"Filtering" denote training without/with the camera encoder/data filtering.

| Method | DINO-Score (↑) | CLIP-Score (↑) | MPJPE (↓) | MRRPE (↓) | FVD (↓) | LPIPS (↓) |
|---|---|---|---|---|---|---|
| Baseline | 48.72 | 62.13 | 402.31 | 370.27 | 425.15 | 0.1570 |
| + Pretrain | 53.81 | 71.29 | 280.51 | 255.63 | 328.41 | 0.1265 |
| + Pretrain&ControlNet | 54.28 | 72.31 | 264.10 | 241.91 | 314.27 | 0.1228 |
| + Pretrain&Entangled | 55.24 | 73.54 | 258.88 | 234.16 | 308.12 | 0.1174 |
| + Pretrain&PMI (No Camera) | 57.68 | 76.79 | 201.34 | 215.89 | 278.24 | 0.1017 |
| + Pretrain&PMI | 59.57 | 78.56 | 170.01 | 191.43 | 266.73 | 0.0912 |
| + Pretrain&PMI&Filtering | 61.74 | 81.11 | 153.17 | 178.64 | 249.33 | 0.0850 |
| + Pretrain&PMI&Filtering&Recon (No Adapter) | 60.13 | 78.67 | 191.22 | 196.54 | 263.18 | 0.1023 |
| + Pretrain&PMI&Filtering&Recon (DUSt3R) | 64.02 | 84.41 | 140.16 | 167.23 | 251.38 | 0.0788 |
| PlayerOne (ours) | **64.87** | **85.79** | **137.91** | **162.74** | **247.97** | **0.0761** |

formation of a genuine connection with the simulated world. In contrast to these limitations, our approach enables freeform motion control for users, enhancing their interactive experience.

**Why separate processing contributes to precise alignment?** Splitting the latent representation into three separate parts—head, hands, and body/feet—enables the model to more effectively capture the unique and often semi-independent motion characteristics of each region. Human body motion is inherently high-dimensional and highly articulated, with different parts frequently moving independently or exhibiting distinct dynamics. Encoding all motion information into a single, entangled latent can cause fine-grained cues—especially from subtle or fast-moving regions like hands or head—to be overshadowed by larger body movements, making precise alignment difficult. By disentangling the motion into part-specific latents, each body region's movement can be independently modeled and aligned with the corresponding input. This modular approach allows the model to better capture local nuances, improves compositionality, and enhances control over complex motions, resulting in higher motion fidelity and visual quality.

**Latency between motion input and video output.** After distillation, the measured end-to-end inference latency is approximately 119 milliseconds per generated-frame, corresponding to about 8.4 FPS. This latency is measured from the moment the motion input is received to the generation of the corresponding video frame, and includes all major processing steps.

**Details of distillation.** We strictly follow the Causvid distillation procedure in our implementation. During the distillation stage, we use the entire SMPL-Video paired dataset from the second phase of training. Specifically, we first generate 8,000 ODE pairs and train the student for 30,000 iterations using AdamW with a learning rate of $5 \times 10^{-6}$. We then continue training with our asymmetric

DMD loss for another 30,000 iterations using AdamW with a learning rate of $2 \times 10^{-6}$. A guidance scale of 3.5 is used, and we adopt the two time-scale update rule from DMD2 with a ratio of 5. The entire distillation process takes about 5 days on 32 A800 GPUs.

**Details of training pipeline.** We clarify that the *world/scene consistency* mechanism is *not* removed at inference. During training, a CUT3R-like reconstructor is used *only* to produce ground-truth point-map sequences corresponding to each video frame; these point maps supervise our scene-frame reconstruction loss and are further encoded to build noised point-map latents. In both training and inference, the *model inputs* are always the initial egocentric frame and the human-motion sequence, while the *prediction targets* are the video frames together with the point-map sequence.

Concretely, let the raw video be $V \in \mathbb{R}^{B \times K \times 3 \times H \times W}$ and its point maps $P \in \mathbb{R}^{B \times K \times N \times 3}$. After VAE encoding, the video becomes $z_{\text{video}}^0 \in \mathbb{R}^{B \times k \times c \times h \times w}$ with $h = H/8$, $w = W/8$, $k = K/4$. A 3D encoder plus an adapter maps the point maps to $z_{\text{point}} \in \mathbb{R}^{B \times k \times 64 \times h \times w}$, matching the video latent's spatio-temporal shape. Both $z_{\text{video}}$ and $z_{\text{point}}$ are noised to obtain $z'_{\text{video}}$ and $z'_{\text{point}}$ for denoising training. The motion sequence, passed through our motion encoder, yields $z_{\text{motion}} \in \mathbb{R}^{B \times k \times 3 \times h \times w}$. The first-frame latent $z_{\text{first}} \in \mathbb{R}^{B \times 1 \times c \times h \times w}$ is repeated $k$ times so that $z_{\text{first}}^k \in \mathbb{R}^{B \times k \times c \times h \times w}$. We then concatenate along channels:

$$z_{\text{input}} = \text{concat}\left( z_{\text{first}}^k, z_{\text{motion}}, z'_{\text{video}}, z'_{\text{point}} \right) \in \mathbb{R}^{B \times k \times (3+64+2c) \times h \times w}.$$

A DiT backbone denoises $z_{\text{input}}$ into $z_{\text{output}}$ (same shape). Losses are computed between the predicted $(z_{\text{video}}^*, z_{\text{point}}^*)$ and the ground-truth $(z_{\text{video}}, z_{\text{point}})$ latents via an MSE objective.

**Inference.** At test time there are no ground-truth point maps and thus no 3D point-map encoder is required. Instead, the network *jointly predicts* the video and its point-map latents autoregressively: both streams are initialized from noise and co-denoised frame by frame, conditioned on the initial frame and the motion sequence. Because the video and point-map streams are modeled with identical spatio-temporal layouts and are denoised in lockstep, the scene-consistency module remains fully active during inference—*no extra inputs are needed and no module is removed*. In short, training uses point maps to supervise and align the latent spaces; inference keeps the joint two-stream denoising (video + point maps) to maintain world consistency throughout generation.

**Superiority over previous video diffusion models.** Our approach advances egocentric video generation and simulation in several complementary ways that prior video diffusion systems do not address. **(i) Scene consistency and world modeling.** Instead of treating frames almost independently—which often causes background flicker, object drift, and abrupt scene changes—our method explicitly reconstructs and maintains a unified 3D world along the entire sequence. We jointly predict the video stream and a point–map stream under an explicit camera representation, so every generated frame is geometrically aligned to an evolving scene state; this yields stable, immersive first-person experiences over long autoregressive roll-outs. **(ii) Accurate and fine-grained first-person control.** Existing methods commonly rely on a single, entangled pose/control vector and therefore cannot model the different roles of head, hands, and feet in egocentric interaction. By employing part-disentangled motion injection, our system exposes independent, precise controls for head orientation (viewpoint/world alignment), hand motion (object interaction), and foot placement (navigation realism), enabling faithful user-driven control and correct coupling between the actor and the rendered environment. **(iii) Efficient and real-time generation.** Rather than invoking heavy 3D encoders at inference, we align modalities in the latent space using lightweight adapters inside the scene–frame reconstruction pathway, and we deploy a distilled backbone for decoding. This design removes bulky inference-time modules and substantially reduces compute/memory cost, enabling high-quality egocentric video at interactive rates suitable for VR/AR and gaming scenarios. **(iv) Comprehensive validation.** We adapt several strong video-diffusion baselines to the egocentric setting under a unified protocol and evaluate them on a reconstructed 100-video benchmark and an extended 24 s horizon. Across perceptual (DINO/CLIP), geometric (MPJPE/MRRPE), and temporal/fidelity (FVD/LPIPS) metrics, our model consistently outperforms adapted baselines, confirming that the gains come from principled world modeling and fine-grained control rather than from model size alone. Collectively, these ingredients make PlayerOne a practical, accurate, and controllable egocentric world simulator, rather than a generic short-clip generator.

**Limitation analysis on our model structure.** From the view of the model, one current limitation of our model is the lack of explicit physical interaction modeling between the human body and the

environment. Structurally, our framework conditions video generation on SMPL motion sequences and a static point cloud reconstruction of the scene, but does not incorporate physical constraints or interaction modules such as collision detection, physics engines, or contact reasoning. Consequently, our model may generate unrealistic results, such as hands or body parts penetrating objects, or a lack of proper occlusion and collision feedback. We acknowledge this limitation and consider it a promising direction for future work.

**More visualization results.**    Here we provide more simulated results in Fig. 11 and Fig. 12. In terms of first-person action alignment and world consistency, we have achieved outstanding results in both game and real-world scenarios. Additionally, we selected highly dynamic settings, such as driving scenes, where our method successfully models the world with high accuracy while maintaining excellent video fluidity. More visualization results can be referred in the submitted video.

**Visualization of the scene reconstruction.**    This section presents a comprehensive visualization of the reconstructed scenes using our PlayerOne. As illustrated in Fig. 9, our approach adeptly reconstructs both scenes and video frames through a progressive methodology. This ensures not only inter-frame coherence but also overarching scene consistency across a diverse array of scenarios. Specifically, the method seamlessly integrates temporal and spatial elements to maintain visual congruity, even in complex environments. The robustness of our technique is further reflected in its capacity to adapt to varying scene dynamics and compositions, thereby offering a reliable framework for generating high-quality, consistent video outputs. Through these visualizations, the effectiveness of our PlayerOnein achieving smooth and coherent reconstructions is clearly demonstrated, highlighting its potential applications in advanced graphical simulations and interactive environments.

**Visual and quantitative difference with the "No Adapter" and "No Recon" setting.**    The large performance gap for 'no adapter' comes from a mismatch in the feature spaces: our point map encoder is fixed, and its latent space is not directly aligned with the VAE-encoded video latent space. When these are simply concatenated without a dedicated adapter, the model struggles to reconcile the different statistics and semantics between the two types of latents, resulting in degraded video quality and scene breakdown. Our adapter module learns to map point map features into the appropriate latent space for effective fusion with video latents, thus improving both fidelity and scene consistency. This design is crucial for effective information transfer between geometry and appearance. For 'no recon', it is primarily due to two reasons:

- Strong Pretraining: Our model is first pretrained on a large-scale egocentric video dataset. This pretraining stage may equips the model with a weak ability to model scene dynamics and maintain a certain level of spatial consistency, even before introducing explicit scene-frame reconstruction.
- Lack of Direct Metrics: Most commonly used metrics (e.g., DINO, CLIP, FVD, LPIPS) are not designed to specifically measure scene (world) consistency across frames. These metrics primarily evaluate frame-wise visual similarity, semantics, or overall diversity, but do not directly capture temporal geometric consistency, which is the core advantage of our reconstruction module.

**Definition of evaluation criteria.**    For user study, we evaluate the video quality from four views:

- Quality: Defined as the overall visual harmony and realism of the video, regardless of whether the generated content is perfectly true to the input. Annotators were asked: "Does the generated video look natural, coherent, and free from jarring artifacts?"
- Smoothness: Measures how consistently the motion flows across frames, without abrupt transitions or temporal artifacts. Annotators were prompted: "Are the motions and camera transitions fluid and continuous throughout the video?"
- Fidelity: Refers to the preservation of the subject's appearance (identity, clothing, background) and the absence of distortions or glitches. Annotators were asked: "Does the person look like the initial frame and is there minimal distortion?"
- Alignment: Defined as how closely the generated motion matches the intended action described in the motion condition. For this criterion, annotators viewed both the generated

video and the motion condition signal (stick-figure or pose sequence) side-by-side for all 100 videos, and rated: "Does the generated video accurately follow the given pose/motion signal in timing, type of action, and spatial positioning?"

# B  Broader Impact

The proposed PlayerOne for video generation, designed to facilitate freeform human motion control within environments created from user-provided images while producing world-consistent videos, demonstrates considerable potential across diverse domains. It is particularly adept at generating engaging and dynamic educational content, thereby fostering experiential learning through interactive simulations. Moreover, the model optimizes the production of high-quality, consistent visual content for films, television, and online media, dramatically reducing both production time and costs. It also enables the creation of interactive narratives, allowing users to influence the storyline through their interactions within the generated environments, thus enhancing user engagement and narrative immersion. Beyond these applications, the world model serves as a valuable tool for research on human behavior and interactions within controlled virtual settings, offering insights for fields such as psychology, sociology, and human-computer interaction. By integrating these capabilities, the proposed world model not only amplifies existing applications but also paves the way for novel research and development, significantly contributing to technological progress and societal advancement.

# C  Limitations & Discussion

While significant strides have been made in egocentric interaction and coherent world modeling, certain limitations persist. Despite the compelling outcomes, the performance in game scenarios is somewhat diminished compared to realistic scenarios, likely due to the disproportionate amount of realistic training data available. Moreover, in highly dynamic scenes, predictions may falter, reflecting the constraints inherent in the current base model. Future research endeavors could potentially overcome these challenges by investigating novel action representations, incorporating an expanded dataset for game scenarios, and adopting a more robust base model.

# D  License of assets

**Datasets**  (Apache 2.0 License) Nymeria [21]/FT-HID [11]/EgoExo-Fitness [18] (Creative Commons Attribution 4.0 International), EgoExo4D [8]/Egovid-5M [33](MIT License).

**Codes**  The official repository of Aether [28] (MIT License), the official repository of Cosmos [1] (Apache 2.0 License), the official repository of Wan2.1 [30] (Apache 2.0 License).

