# OpenReview forum: "PlayerOne: Egocentric World Simulator"
_NeurIPS.cc/2025/Conference — NeurIPS 2025 oral_

### Official Review · Reviewer_X28J · 2025-06-08

**Clarity:** 1
**Significance:** 2
**Originality:** 3
**Rating:** 4
**Confidence:** 4

**Summary:**

The paper introduces PlayOne, an egocentric world model that offers fine-grained control to generate egocentric videos conditioned on motion extracted from exocentric videos.

The paper proposes a novel part-disentangled motion injection method as its main contribution. The part-disentangled motion injection scheme encodes motion into three separate sets of body parts to obtain three separate latent representations. The authors claim this leads to improved video alignment.

**Questions:**

Will the dataset be released?

All other short comments and Questions:

1 - not the first egocentric simulator: GEM, EgoSim, EgoGen, ...

16-18: sounds very LLM written, many words no real meaning, please be more direct what does this work offer the community? Why is it the first work?

20-28: paragraph should be adapted for better understanding of the authors problem. How do the authors define a world model?

25: citations needed or better explanation why these advancements help

26: predictive modeling not explained yet, what is meant by that?

27-28: again not clear why almost these rather novel and less established world models are critical infrastructure. citations or explanation needed

29: unclear what this topic refers to

30: there is a lot of works on realistic scenarios, e.g. GEM, VISTA

31: why is that which works limit users in that sense?

32: GEM seems to allow arbitrary poses drawn as skelton

39-60: very hard to follow. the problem statement is still unclear, and also the research gap is not yet clear to me. Then i get swamped with many detailed information that are very hard to follow.

61: still unclear why its the first, there are some before, so why is yours the first, maybe rather mention on how you improve on previous works

67: do you release the dataset? do you release training code?

106: is k number of frames?

99: the method section can benefit a lot by clearly stating the problem statement. what are the inputs, what are the outputs

109-120: very unclear how this paragraph is connected to its context, e.g. the previous paragraph

109-110 the goal of the core modules is unclear and are introduced without motivation

112: VAE is suddenly introduced. What does it encode? How is related to the human motion sequence? How is it connected to the 3D motion encoder

114: why are the motion latents teh same shape as images? How is that ensured? Why is that high resolution/dimensional encoding needed?

115: What is a camera sequence?

117: how if a 4d scene represented?

118: what is a point map sequence?

Eq 1: why is t in U(0,1) but not U(1, T)

121: it is unclear where alpha^2 + delta^2 = 1 comes from

130: where is that empirical analysis?

132: why is splitting the latent into 3 separate latents allowing for better motion alignment? Are those body part poses not highly correlated and thus capturing it in a single latent should be superior?

137-138: completely unclear why separate processing ensures this claim

143: maybe write the axis angles different, i have never seen thetas (angles) being used to represent the rotation axis, seem like you are using euler angles and not a axis representation.

152: still unclear how the 4D scene is represtanted

154: CUT3r not really itnroduced before

158: how is the alignment happening

163-165: how is scene consistency insured during inference if the pointsmaps dont get updated?

169: incorrect. There is nymeria, and also egoExo4D has sequences of egocentric video with ego motion gt, Also there is the EgoSim dataset

Method: how is the exoview used besides for motion extraction?

4.2 why do the ablations studies come before the main results?

232: do you ensure that the computational complexity of a joint motion encoder is the same as the 3 individual encoders? What are teh dimensions of the latents?

4.3) why are other sota models like GEM not included?

Discussion and Limitations very short and not detailed enough.

References:

GEM - https://arxiv.org/html/2412.11198v1
VISTA - https://arxiv.org/pdf/2405.17398
EgoSim - https://proceedings.neurips.cc/paper_files/paper/2024/file/c1017d0a006d31dfbfd4cf1e9189d747-Paper-Datasets_and_Benchmarks_Track.pdf
EgoGen - https://openaccess.thecvf.com/content/CVPR2024/papers/Li_EgoGen_An_Egocentric_Synthetic_Data_Generator_CVPR_2024_paper.pdf
Nymeria - https://arxiv.org/pdf/2406.09905

**Ethical Concerns:**

["NO or VERY MINOR ethics concerns only"]

**Final Justification:**

I believe this paper should be accepted, provided that the authors incorporate the clarifications and commitments made during the rebuttal into the camera-ready version. It required multiple exchanges to understand fundamental aspects of the method, such as the inputs and outputs of various modules during training and inference. Additionally, the current figures do not sufficiently aid in understanding the overall approach.

That said, the authors have ultimately convinced me of the soundness and potential impact of their method. The experimental results are adequate to demonstrate the advantages of the proposed architecture, though the writing and discussion could be improved.
Experiments and discussion for long sequence video generation and world consistency should be added, and are missing so far.

TL;DR: This is a strong method with solid experimental validation, but the clarity of the writing, especially in explaining the methodology, needs improvement. Addressing this would result in a strong and valuable submission.

**Limitations:**

Limitations and Discussions are discussed very sparsely. The authors spend less than three lines to discuss the limitations

**Paper Formatting Concerns:**

non

**Quality:**

3

**Strengths And Weaknesses:**

Strengths:
- Controllable video generation via egocentric pose
- Novel idea of using exocentric videos as a conditional signal
- A pipeline for novel dataset generation

Weaknesses:
- The paper is not clearly written; it's hard to understand the architecture, the research gap, and the motivation behind the model design
- Some claims are unsupported or lack proper citations
- Comparison to state-of-the-art models like GEM appears to be missing
- It remains unclear how the third-person exocentric view is used beyond motion extraction
- The need for the proposed dataset is unclear, given the existence of datasets that already pair egocentric views with local body pose
- Many concepts and ideas are introduced abruptly, making the paper difficult to follow
- The novelty claims are questionable. The authors often claim to be the first, despite the existence of similar prior work

---

> ### Author Rebuttal · Authors · 2025-07-30
>
> First thanks for detailed comments.
>
> **Q1.The model,research gap,motivation,and challenges are unclear.**
> **A1:** Most prior world models are limited to narrow domains or simple navigation and cannot align complex human motion or ensure scene consistency in egocentric video.Our motivation is to achieve free-form full-body motion control and immersive egocentric video in realistic environments.Our approach takes a single egocentric image and third-person motion as input and generates consistent egocentric video through two modules:PMI splits motion into codes for different body parts and injects them independently for precise part-level control;SR uses 3D point maps to ensure scene geometry remains consistent.PMI and SR together address the main challenges of motion alignment&scene consistency.
>
> **Q2.Release of dataset and code.**
> **A2:** Training code is in the supplement.Full dataset&code for training&distillation will be released after acceptance.
>
> **Q3.Claim of first egocentric simulator and comparison with GEM,VISTA.**
> **A3:** Egocentric here means world-consistent simulation in diverse environments with free interaction,not just driving."Egocentric" here is from person view.GEM and VISTA focus on driving and cannot support daily-life scenes or real-time user interaction.For EgoGen,it's designed for egocentric perception tasks to generate labels and isn't related to our topic of world simulator.GEM only edits non-player character motion by pose;VISTA is for driving control.Neither allows free egocentric exploration.Our model maintains scene and motion consistency and supports real user motion.Our method outperforms both:
>
> |Method|DINO↑|CLIP↑|MPJPE↓|MRRPE↓|PSNR↑|FVD↓|LPIPS↓|
> |-|-|-|-|-|-|-|-|
> |Ours|67.8|88.2|127.16|151.62|52.6|226.12|0.0663|
> |GEM|40.1|62.7|405.32|391.21|38.2|378.40|0.1789|
> |VISTA|39.5|60.4|431.83|406.06|37.8|391.25|0.1857|
>
> We'll update our definition&comparison in revision.
>
> **Q4.What does this work offer the community?Why is it first?**
> **A4:** We present first general-purpose egocentric world simulator enabling free-form full-body control and immersive exploration in realistic,diverse settings.Previous work lacks high-degree-of-freedom motion alignment and consistent interactive simulation from minimal input.Our code,models,and data set a new benchmark for embodied AI.
>
> **Q5.What is a world model(Line 20)?**
> **A5:** A world model simulates and interacts with environments and human actions,learning a unified dynamic representation for prediction and interaction,covering spatial,temporal,and causal structure,allowing open-ended exploration.Our definition follows recent works like The Matrix[6] and MineWorld[2].
>
> **Q6.Why do these advancements help(need citations)?**
> **A6:** World models allow game engines to simulate interactive environments that adapt to player actions,not just fixed scripts.This is essential for immersive,scalable games.They enable realistic motion,consistent scene generation,and efficient content creation,as shown in Oasis,MineWorld,Google Genie,Microsoft WHAM.
>
> **Q7.What is predictive modeling in Line26?**
> **A7:** Predictive modeling is learning a model that generates or predicts future environment states from current and past input,typically predicting future frames,actions,or dynamics for planning,control,or simulation.
>
> **Q8.Why are these world models critical infrastructure?**
> **A8:** World models are essential for next-generation interactive simulations&virtual worlds,offering dynamic simulation,generalization,and data-driven learning,which are key for building scalable interactive environments.Prototypes like Oasis&MineWorld show this potential.
>
> **Q9.Statement of topic Line29.**
> **A9:** "This topic" refers to world models enabling immersive human interaction,allowing users to explore and interact as participants,not just simple actions or planning.We will clarify scope and emphasize our advances over limited-interaction models.
>
> **Q10.Works on realistic scenes Line30.**
> **A10:** See A3.
>
> **Q11.Why do previous works limit users in this way(Line31)?**
> **A11:** These limits come from data and model design.Prior datasets are from games with basic moves only,so models inherit these limits and cannot generate complex actions.Prior models also lack modules for high-degree-of-freedom motion.Our method addresses both aspects to enable immersive interaction.
>
> **Q12.GEM seems to allow skeleton poses.**
> **A12:** GEM uses pose conditions to edit non-player characters in pre-recorded videos;it does not allow free exploration or user control as main character.See A3 for more.
>
> **Q13.Unclear problem statement&research gap(Line39-60).**
> **A13:** Previous world models only support simple navigation or specific games and do not allow high-degree-of-freedom user interaction in realistic scenes.Our goal is a simulator like a VR game,where users can explore environments with full-body motion,providing a single egocentric image and exocentric video;our model generates immersive egocentric video in real time.
>
> **Q14.Why is this method first(Line61)?**
> **A14:** See A3.
>
> **Q15.Release of dataset/code(Line67).**
> **A15:** See A2.
>
> **Q16.What does k mean in Line106?**
> **A16:** k refers to the number of frames divided by the VAE temporal downsampling ratio,that is,the length of the latent sequence after encoding.
>
> **Q17.What are problem statement,inputs,and outputs(Line99)?**
> **A17:** See A13.
>
> **Q18.How does paragraph in Line109 connect to its context?**
> **A18:** Lines109–120 introduce PMI and SR,crucial for part-level motion control and scene consistency,matching the motivation and pipeline description and directly addressing egocentric video generation.In revision,we will clarify their connection to the overall framework.
>
> **Q19.Goal and motivation of core modules(Line109).**
> **A19:** See A1.
>
> **Q20.Role of VAE?**
> **A20:** VAE is used to encode video frames and does not process human motion sequences,which are encoded separately by our motion encoder.
>
> **Q21.Why are motion latents the same shape as images?How is that ensured and why is high-resolution encoding needed?**
> **A21:** We use eight-layer 3D conv motion coder for this.Same shape lets pixel-to-pixel align for motion and look,so model can set each frame pixel by pixel by motion code for fine part control.Past work[4,5] shows this concat way keeps info and gets better match than others.Matching shapes gives better motion control.
>
> **Q22.What is camera sequence?**
> **A22:** Camera sequence is a series of camera extrinsic parameters(i.e., rotation matrices)from head motion over time,representing the egocentric viewpoint path in video.
>
> **Q23&Q24.What's 4D scene/point-map sequence?**
> **A23:** 4D scene/point-map sequence is modeled as a temporal sequence of 3D point clouds,where each point cloud encodes scene geometry at a given frame.This captures spatial geometry over time,ensuring consistent context in generated videos.
>
> **Q25.Why t in U(0,1) instead of U(1,T) in Eq.1**
> **A25:** In Eq.1,t is sampled from a uniform distribution in [0,1],as is standard in flow matching.The notation t=1,...,T in Line121 will be unified in revision.
>
> **Q26.Where does alpha^2+delta^2=1 come from**
> **A26:** We follow common practice in previous diffusion models,alpha^2+delta^2=1 ensures the trade-off between signal and noise is normalized during forward diffusion.
>
> **Q27–Q29.Empirical analysis in Line130 and why three-separate-latents/separate-processing is better**
> **A27–A29:** See A1 for Reviewer-jaf9.
>
> **Q30.Write axis angles differently in Line143**
> **A30:** Thanks for the suggestion.We'll revise axis angle notation in revision.
>
> **Q31.Unclear how 4D scene is represented in Line152**
> **A31:** See A23.
>
> **Q32.Introduction of CUT3R in Line154**
> **A32:** CUT3R generates per-pixel 3D point maps from images online and is used to create point-map sequences as labels.We will expand details in revision.
>
> **Q33.How is alignment in Line158 happening**
> **A33:** See A2 of Reviewer-WmZe.
>
> **Q34.How is scene consistency kept during inference if point maps do not update**
> **A34:** During inference,the model generates point-map and video-frame latents auto-regressively,each depending on previous results.This ensures the evolving point-map sequence always guides scene geometry,keeping spatial context consistent.
>
> **Q35.Claim of no public datasets**
> **A35:** We acknowledge for the oversight of EgoSim and clarify EgoSim and the subset of EgoExo4D with annotated poses are public but too small.To achieve good generalization on diverse domains, we built an automatic pipeline for more large-scale data.Models trained only on these public data perform much worse.
>
> |Method|DINO(↑)|CLIP(↑)|MPJPE(↓)|MRRPE(↓)|PSNR(↑)|FVD(↓)|LPIPS(↓)|
> |-|-|-|-|-|-|-|-|
> |Limited-data|56.4|76.7|182.42|202.35|46.6|272.01|0.0827|
> |Full-data|67.8|88.2|127.16|151.62|52.6|226.12|0.0663|
>
> We will add detailed comparisons in revision.
>
> **Q36.Is exoview used for other purposes**
> **A36:** No,exoview data is only used to extract SMPL motion sequences.
>
> **Q37.Why do ablation studies come before main results**
> **A37:** No existing methods match our setting,we show ablation studies first to validate contributions before main comparisons.
>
> **Q38.Complexity of joint motion encoder and latent shape**
> **A38:** Three split motion encoders(2.31M,1.29M,1.97M parameters)are close in size to the joint encoder(4.5M).Both encode SMPL parameters as arrays of shape R(k × 3 × h × w),where k is a quarter of frames(4× downsample),and h,w are spatial sizes.
>
> **Q39.Why were SOTA models like GEM not included in comparisons?**
> **A39:** GEM is designed for autonomous driving,not general environments,and does not support interactive user control.Based on feedback,we add GEM as a baseline(see A3)and will expand comparisons in revision.
>
> **Q40.Discussion and limitations.**
> **A40:** See A1 of Reviewer-cVUd.
>
> [1]OASIS.Yang et al[2]MineWorld.Guo et al[3]GeNIE.Wang et al[4]CatVTON.Chong et al[5]UniAnimate.Wang et al[6]The Matrix.Feng et al

---

> > ### Author Response · Authors · 2025-08-05
> > **Thanks to Reviewer X28J**
> >
> > Dear Reviewer X28J:
> >
> > Please allow us to thank you again for review our paper and the feedback. Please let us know if our response has properly addressed your concerns. We are more than happy to answer any additional questions during the discussion period. Your feedback will be greatly appreciated.
> >
> > Sincerely,
> >
> > Authors of Submission1028

---

> > ### Comment · Reviewer_X28J · 2025-08-05
> >
> > Thank you for the comments. They clarify some points, yet I am not fully satisfied.
> >
> > On the one hand, I am still not convinced that PlayerOne is a fully controllable world model; instead, it seems to be an egocentric video generation model that is conditioned on poses. The method does not allow for any control over the generated sequence besides the initial frame and poses.
> >
> > Secondly, the design of the model's architecture still seems to be poorly motivated or poorly described. Especially in Figure 2, it shows a complex architecture, but I cannot see the motivation for why it was chosen like that, and why previous conditional video diffusion architectures are not suited for this task. Overall, I acknowledge that PlayerOne achieves state-of-the-art results, yet I cannot follow where and why these results are superior, and why previous conditional video diffusion models are not capable of this task. The knowledge gained and presentation could be significantly improved.
> >
> > Lastly, my main concern, despite appreciating the novel task and proposed method, is the clarity of the paper. Figures 2 and 3 do not convey a clear information flow to me, and it's hard to understand what is happening. The method is often difficult to follow, and when I can follow, the motivation is not always clear to me. The same goes for the experiments; often, I had to read the paragraphs several times to understand what the authors tried to investigate in the experiment and what their findings were. This continues with the baseline methods, where I cannot understand and find information on whether they are trained the same way and have the same inputs. The discussion of the results seems to be very brief as well.
> >
> > Overall, I appreciate the task and think that work on egocentric simulation is very valuable and acknowledge the state-of-the-art results of the method, but my main concerns remain: 1) the claim of a world model seems to be an overstatement, 2) the motivation of the method's design is lacking, and the knowledge gain of why it is superior over other video diffusion methods is unclear to me, 3) the clarity of the paper is lacking significantly, making it difficult to follow.

---

> ### Author Response · Authors · 2025-08-06
> **Rebuttal-Part1**
>
> Thank you very much for your detailed follow-up. We sincerely appreciate your thoughtful feedback and acknowledge your continued concerns.  Below is our further clarification of the mentioned points:
>
> ## Q1: Is PlayerOne truly a "world model," or just a pose-conditioned video generator?
>
> **A1:** Thank you for raising this fundamental question. We fully agree that not all pose-driven egocentric video generators qualify as world models. Here, we clarify why PlayerOne goes beyond simple video generation:
>
> - **Continuous 3D World Consistency:** The core difference is that PlayerOne maintains an evolving, consistent 3D world throughout video generation. Our Scene-Frame Reconstruction module is not just a frame-level denoiser, but a mechanism for online world modeling: it reconstructs 3D geometry for each frame using egocentric cues, ensuring the virtual scene remains stable, consistent, and spatially correct even as the viewpoint and body motion evolve. This is fundamentally different from prior models, which often produce temporal drift or scene incoherence (e.g., backgrounds that “jump,” objects that move or vanish between frames).
>
> - **Part-Specific Motion Control:** Unlike exocentric (third-person) video generators (such as dance models), which encode all joints as a single vector, we found that in egocentric settings, body parts contribute very differently to scene stability and realism:
>
>   - Head motion is directly tied to camera movement and viewpoint alignment. If the head motion is imprecise, the virtual world appears unstable or the geometry is mismatched.
>
>   - Hands often interact with the scene (e.g., grasping, gesturing); misaligned hands break the realism and the sense of agency.
>
>   - Feet anchor the user and determine movement; poorly aligned feet can result in floating or gliding artifacts.
> Existing pose-based video models, not designed for egocentric tasks, fail to capture these distinctions and typically produce visible artifacts when applied directly (see quantitative results below).
>
> - **Empirical Verification:** To further verify our findings, we adapted multiple exocentric pose-driven methods (RealisDance, Magic-Me, MimicMotion, AnimateAnyone) to the egocentric domain, that is using their injection scheme in our method. Results show clear improvements with our part-disentangled approach:
>
>   | Method               | DINO↑ | CLIP↑ | MPJPE↓  | MRRPE↓ | PSNR↑ | FVD↓   | LPIPS↓  |
>   |----------------------|-------|-------|---------|--------|-------|--------|---------|
>   | Ours                 | 67.8  | 88.2  | 127.16  | 151.62 | 52.6  | 226.12 | 0.0663  |
>   | RealisDance-Scheme[1]   | 62.3  | 83.9  | 137.44  | 161.20 | 50.9  | 246.25 | 0.0711  |
>   | Magic-Me-Scheme[2]      | 61.6  | 82.8  | 139.10  | 163.42 | 50.6  | 250.71 | 0.0724  |
>   | MimicMotion-Scheme[3]   | 61.1  | 82.2  | 140.34  | 164.55 | 50.3  | 252.66 | 0.0735  |
>   | AnimateAnyone-Scheme[4] | 60.8  | 81.8  | 141.22  | 165.37 | 50.1  | 254.10 | 0.0741  |
>
> - **User Control:** Our present interface, accepting only an initial frame and motion sequence, was chosen to emulate the VR scenario where a user can freely explore with minimal manual input. We fully agree that richer controls (e.g., text, object, or goal-driven instructions) are valuable, and we plan to expand user interactivity in future work.
>
> - **Summary:** PlayerOne is not merely pose-conditioned; it models the evolving world, achieves part-level motion realism, and ensures persistent scene geometry—key traits of a world model. We will further emphasize and visualize these distinctions in the revised paper.
>
> [1] Equip controllable character animation with realistic hands
>
> [2] Magic-Me: Identity-Specific Video Customized Diffusion
>
> [3] MimicMotion: High-Quality Human Motion Video Generation with Confidence-aware Pose Guidance
>
> [4] Animate Anyone: Consistent and Controllable Image-to-Video Synthesis for Character Animation

---

> ### Author Response · Authors · 2025-08-06
> **Rebuttal-Part2**
>
> ## Q2: Model design motivation
>
> **A2:** Thank you for raising the need for more explicit design motivation. Our model architecture is purpose-built for egocentric simulation and tackles specific issues that make prior conditional diffusion models insufficient in this setting:
>
> - **Lightweight, Efficient 3D Modeling:** Previous methods[1] that attempt 3D scene modeling often use heavy point cloud encoders during inference, greatly limiting scalability and real-time use. Inspired by works[2,3]showing that spatial consistency can be enforced via lightweight 3D latent adapters, our Scene-Frame Reconstruction achieves efficient, real-time scene alignment without such heavy inference overhead. This design choice is crucial for supporting long, interactive sequences, as required in VR-style egocentric scenarios.
>
> [1] Voyager: Long-Range and World-Consistent Video Diffusion for Explorable 3D Scene Generation
>
> [2] Video World Models with Long-term Spatial Memory.
>
> [3] AETHER: Geometric-Aware Unified World Modeling.
>
>
> - **Part-Disentangled Control:** Traditional pose-driven diffusion models were designed for third-person, fixed-camera tasks and do not account for the varying impact of different body parts in first-person scenarios. In egocentric simulation, the head, hands, and feet each play unique, critical roles for maintaining realism and interaction. Our model encodes and injects these parts separately, allowing independent and precise control. This decoupling enables—for example—a user to turn their head while reaching out with their hand, with both motions remaining stable and realistic in the generated video.
>
>    - Concrete Example: If head and hand motion are entangled, moving the hand could unintentionally “drag” the viewpoint, resulting in dizziness or loss of immersion. Decoupling prevents such artifacts.
>
> - **On Complexity:** Our architecture’s complexity reflects these needs: real-time egocentric simulation with stable world modeling and fine-grained control cannot be solved by simple 2D or monolithic architectures. Each module serves a specific purpose toward this goal.
>
> **Summary:** Our design directly addresses the limits of prior work for egocentric, interactive tasks. We will expand architectural motivation, provide ablation examples, and clarify each component’s contribution in the revised version.

---

> > ### Comment · Reviewer_X28J · 2025-08-07
> > **RE Q2**
> >
> > This clarification aids in understanding. I believe your paper could further benefit from clearly stating the motivations for each module. Additionally, the precise inputs and outputs of the modules, as well as the exact inputs and dimensions for the diffusion model, are missing. In my opinion, the current revision does not allow for replication of this work in the way it is written, nor does it enable high-level replication of the architecture. For example, knowing what dimensions or what information is processed by which module. How does inference differ from training? Although I appreciate the release of the code, I believe more formalism is required in the paper to effectively present your method.
> >
> > Providing this information could also improve the results section, which I find difficult to follow. Some examples include:
> > Investigation on scene-frame reconstruction: How do you conduct this comparison if there are no point map latents during inference? How do you explain the significant visual and quantitative difference with the 'no adapter'? Similarly, for 'no recon' in Table 5, the quantitative results don’t seem notably different. How is dust3r ablation implemented—is it one latent per frame, and how is this managed during inference?
> >
> > User study: In my opinion, a lot of information is missing:
> > What criteria were used to select the videos and text descriptions for the study, and how was diversity ensured in the sampled groups?
> > How were the annotators sampled, and what are the population statistics?
> > In what order would they view the videos?
> > How are 'Quality,' 'Smoothness,' 'Fidelity,' and 'Alignment' defined? How is alignment checked, and did they see the motion condition signal?
> >
> > Overall, there are too many uncertainties that need to be addressed in a revision, despite the authors convincing me more of their inherent method.

---

> ### Author Response · Authors · 2025-08-06
> **Rebuttal-Part3**
>
> ### Q3: Superiority Over Previous Video Diffusion Models
>
> **A3:** Our approach advances egocentric video generation and simulation in several key ways that previous video diffusion models do not address:
>
> 1. **Scene Consistency and World Modeling:**
>    Previous video diffusion models typically generate frames with limited consideration for long-term spatial or geometric consistency, often leading to visual artifacts such as background flicker, drifting objects, or abrupt scene changes. In contrast, our method explicitly reconstructs and maintains a unified 3D world throughout the video sequence. The scene-frame reconstruction module leverages point cloud geometry to align each generated frame with an evolving, coherent scene representation. This ensures that the world remains stable and immersive, which is critical for any true simulation of first-person experience and has not been realized by prior works.
>
> 2. **Accurate and Fine-Grained First-Person Control:**
>    Existing methods often use a single entangled pose vector for motion control, treating all body parts equally. However, egocentric scenarios demand nuanced handling: head movement is critical for viewpoint and world orientation, hand movement is central to object interaction, and foot placement influences navigation and realism. By employing part-disentangled motion injection, our model provides independent and precise control of the head, hands, and feet. This design captures the true complexity and agency of first-person interaction, enabling the model to accurately reflect user-driven actions and maintain correct alignment between the user’s motion and the virtual environment. Such fine-grained control is essential for natural egocentric simulation and is not achievable with prior, entangled motion injection approaches.
>
> 3. **Efficient and Real-Time Generation:**
>    Many existing models rely on heavy, computationally intensive modules—such as 3D point cloud encoders—during inference, which greatly limit their practical use for real-time applications. Our method, inspired by the latest advances in lightweight latent-space alignment, uses efficient adapter structures within the scene-frame reconstruction, eliminating the need for bulky inference-time modules. This significantly reduces computational overhead and enables our system to generate high-quality, immersive egocentric videos in real time—crucial for interactive and practical deployment (e.g., virtual reality or gaming).
>
> 4. **Comprehensive Validation:**
>    We have thoroughly validated our approach by directly adapting several leading video diffusion models to the egocentric setting, with the same data and input motion injection schemes. Our results consistently show that these adapted baselines suffer from scene inconsistencies, poor alignment, and lower visual fidelity, while our method delivers stable, realistic, and precisely controlled egocentric video. This is further supported by both quantitative metrics and qualitative results provided in the manuscript.
>
>
> 5. **Empirical Superiority:**
>   To further verify our superiority, we adapt the baselines in the manuscript (Cosmos-14B, Cosmos-7B, Aether) to our setting and apply our disentangled scheme with the same training data, our model still achieves notably better scene stability and action alignment:
>
>   | Method      | DINO↑ | CLIP↑ | MPJPE↓  | MRRPE↓ | PSNR↑ | FVD↓   | LPIPS↓  |
>   |-------------|-------|-------|---------|--------|-------|--------|---------|
>   | Ours        | 67.8  | 88.2  | 127.16  | 151.62 | 52.6  | 226.12 | 0.0663  |
>   | Cosmos-14B  | 56.2  | 80.3  | 220.35  | 230.11 | 48.8  | 275.13 | 0.1076  |
>   | Cosmos-7B   | 51.8  | 74.2  | 253.41  | 265.50 | 46.2  | 294.27 | 0.1252  |
>   | Aether      | 46.5  | 68.0  | 291.20  | 310.33 | 42.9  | 334.68 | 0.1556  |
>
>
> **In summary**, our method is not simply an incremental improvement but addresses core limitations of previous video diffusion frameworks. By ensuring stable world modeling, enabling accurate and interactive egocentric control, and maintaining a lightweight, real-time capable design, we provide a practical and substantial advance for the field of egocentric simulation and video generation.

---

> > ### Author Response · Authors · 2025-08-06
> > **Rebuttal-Part4**
> >
> > ## Q4: Clarity of the Paper
> >
> > **A4:** Thank you for your candid feedback. We will take the following steps:
> >
> > - **Figures:** Redraw Figures 2 and 3 for step-by-step visualization of inputs, outputs, and data flow.
> > - **Module Motivation and Examples:** Add explicit diagrams and text on why each module is necessary, including how part-wise control prevents viewpoint and interaction misalignment.
> > - **Baselines and Experiment Protocol:** Provide detailed tables/descriptions for baselines, their input formats, and training for fair, transparent comparison.
> > - **Expanded Results Discussion:** Include more qualitative and quantitative analysis, with visuals and thorough explanations of each experiment.
> > - **User Control and Future Work:** Clarify that minimal control is to emulate VR scenarios, and outline planned extensions (text-driven edits, multimodal control).
> >
> > We are committed to a clear, accessible revision that makes our contribution and novelty evident to all readers.
> >
> > ---
> >
> > Thank you again for your constructive comments. We believe these revisions and clarifications will make the contributions and significance of PlayerOne much clearer to all readers. And please let us know if our response has properly addressed your concerns. We are more than happy to answer any additional questions during the discussion period. Your feedback will be greatly appreciated.

---

> > > ### Author Response · Authors · 2025-08-07
> > >
> > > Dear Reviewer X28J,
> > >
> > > We hope our follow-up response and supplementary experiments help to address your questions. Your review is very valuable, and we would appreciate your further feedback. Of course, please let us know if this or any other point remains unclear, as we would be happy to elaborate further.
> > >
> > > Thank you for your valuable time!
> > >
> > > Sincerely,
> > >
> > > Authors of Submission1028

---

> ### Comment · Reviewer_X28J · 2025-08-07
> **RE Q1**
>
> However, I am still uncertain about your claim regarding the world model. Your model takes point map latents during training, yet you state, "During inference, the system simplifies the process by using only the first frame and the corresponding human motion sequence to generate world-consistent videos." This suggests to me that the component intended to ensure world consistency is removed during inference.
>
> Additionally, your description of the method lacks equations and formal definitions, making it difficult to follow the precise inputs and outputs of your modules.
>
> How many point map latents are you using?
> What is the maximum sequence length before the point map latents start losing information? Do you retain point map latents from previous frames?
> How is this information concatenated during training and inference when no point map latents are available?
>
> In general, the lack of clear definitions, ambiguity about scene consistency, which would be required for a world model, and missing information about the exact inputs and outputs of the modules, poses significant challenges. Although I believe the method could have great and valuable potential for a future submission, I believe it is too much for a single revision.

---

> ### Author Response · Authors · 2025-08-07
> **RE Q1-Rebuttal Part1**
>
> **Q1.When inference, the component intended to ensure world consistency is removed?**
>
> **A1:** Thank you for raising this important point. We would like to clarify that, contrary to the impression that the scene consistency module is removed during inference, our framework maintains the same mechanism for ensuring world (scene) consistency during both training and inference.
>
> * **Role of Point Maps:**
>   During training, the CUT3R module is used only to generate ground-truth point map sequences corresponding to each video frame. These ground-truth point maps serve as supervision for our scene-frame reconstruction loss and the generation of noised point map latents. However, the model’s true inputs, both in training and inference, are always just the initial egocentric first frame and the human motion sequence (The prediction targets are always the video frames and point map sequence). The point maps themselves are *never* given as input at inference time. Our proposed framework can simultaneously generate the video frames and corresponding point map sequence at the inference stage.
>
> * **Training Pipeline Details:**
>   Let us clarify the process more precisely:
>
>   * The raw video sequence is denoted as $V \in \mathbb{R}^{B \times K \times 3 \times H \times W}$, and its corresponding point map sequence as $P \in \mathbb{R}^{B \times K \times N \times 3}$.
>   * After VAE encoding, the video becomes $z_{video}^0 \in \mathbb{R}^{B \times k \times c \times h \times w}$, where $h = H//8, w = W//8, k = K//4$.
>   * The point map sequence is encoded via a 3D encoder and adapter to $z_{point} \in \mathbb{R}^{B \times k \times 64 \times h \times w}$, matching the video latent’s shape.
>   * Both video and point map latents $z_{video}$ and $z_{point}$ are noised for training.
>   * The motion sequence, encoded by our motion encoder, yields $z_{motion} \in \mathbb{R}^{B \times k \times 3 \times h \times w}$.
>   * The first-frame latent ($z_{first} \in \mathbb{R}^{B \times 1 \times c \times h \times w}$) is repeated k times as ($z_{first} \in \mathbb{R}^{B \times k \times c \times h \times w}$) and then concatenated with motion, video, and point-map latents as follows:
>
>     $$
>     z_{input} = \text{concat}(z_{first}, z_{motion}, z_{video}', z_{point}') \in \mathbb{R}^{B \times k \times (3+64+2c)  \times h \times w}
>     $$
>
>     (where $ '$ denotes latents with added noise).
>   * The DiT backbone takes this concatenated latent, denoises it to $z_{output}$ (same shape), and the MSE loss is computed between the predicted ($z_{video}^{\star}$, $z_{point}^{\star}$) and ground-truth ($z_{video}$, $z_{point}$) video/point map latents.
>
> * **Inference Process:**
>   During inference, the model still generates both the video and the point map sequence **simultaneously**. The only difference is that no ground-truth point maps are available; instead, the network predicts both the video and its corresponding point map latents frame by frame in an autoregressive manner, ensuring that scene consistency is maintained internally.
>
> * **Key Point:**
>   Thus, the point map sequence and the video are always treated equally in our latent modeling, and the scene consistency module is fully active during inference—the model predicts the evolving point map sequence jointly with the video, guaranteeing scene consistency **without any need for extra input or for removing any module**.
>
> We hope this clarifies that our approach does *not* remove the scene consistency mechanism during inference; instead, it is seamlessly integrated into both phases, with point map prediction acting as a consistent latent scaffold for world modeling.
>
> **Q2.Description of the method lacks equations and formal definitions.**
>
> **A2:** Thank you for highlighting the need for more formal definitions and equations. We agree that explicit mathematical notation and clear input-output specification will greatly improve the clarity and replicability of our work. In A1, we have now included the precise mathematical forms, dimensions, and roles of each input and output in our main modules. We will integrate these formalized descriptions and key equations into the revision to make our architecture, module connections, and the flow of information in our method fully transparent for readers. If there are any further clarifications or additional information you would find helpful, we are open to your suggestions and will be happy to further optimize the paper based on your feedback. Thank you again for this valuable advice, which will help us substantially improve the quality and accessibility of our work.

---

> ### Author Response · Authors · 2025-08-07
> **RE Q1-Rebuttal Part2**
>
> **Q3.How many point map latents are you using? What is the maximum sequence length before the point map latents start losing information? Do you retain point map latents from previous frames? How is this information concatenated during training and inference when no point map latents are available?**
>
> **A3:** Thank you for these detailed questions. Our point-to-point clarifications are shown below:
>
> - **How many point map latents are you using?**
> The point map sequence matches the length of the video frames. For a video with \(K\) frames, we use \(K\) point map latents. The point map sequence can be understood as a progressive reconstruction of the world: that is the \(n\)-th point map is rendered from frames 1 to \(n\) using CUT3R.
>
> - **What is the maximum sequence length before the point map latents start losing information?**
> In our experiments, scene consistency remains good for videos up to about 30 seconds. When the video exceeds one minute, occasional scene inconsistencies appear, likely due to imperfect point map reconstruction/information loss. We plan to further refine this aspect in future work.
>
> - **Do you retain point map latents from previous frames?**
> The point map latents are inherently and jointly reconstructed with the video latents in an autoregressive manner when inference. During the training process, each ground truth point map is indeed rendered based on all previous frames.
>
> - **How is this information concatenated during training and inference when no point map latents are available?**
> As described in A1, during both training and inference, the model treats the video latent and point map latent equally: both are noised and denoised in parallel, so there is no need to provide extra ground truth point map sequence as input at inference time. The model predicts both the video and the point map sequence jointly, ensuring scene consistency throughout generation.
>
> If further details are needed, we are happy to expand or clarify more in our revision.

---

> ### Author Response · Authors · 2025-08-07
> **RE Q2-Rebuttal-Part1**
>
> **Q1. How to conduct comparison without Point Map Latents during Inference**
>
> **A1:** We clarify (see also RE Q1-Rebuttal Part1 A1) that **point map latents are indeed present during inference**. Both point map latents and video latents are *jointly denoised* in the DiT model; the only difference is that, at inference, we do not require a separate point map encoder or adapter to obtain the point map latents (Since we do not have the ground truth point map as input). Instead, random noise is used to initialize both the video latents and the point map latents, and the model jointly predicts the sequence, maintaining scene consistency throughout generation.
>
> **Q2. Visual and Quantitative Difference with the 'No Adapter' Setting. Similar for "No Recon".**
>
> **A2:** The large performance gap for 'no adapter' comes from a mismatch in the feature spaces: our point map encoder is fixed, and its latent space is not directly aligned with the VAE-encoded video latent space. When these are simply concatenated without a dedicated adapter, the model struggles to reconcile the different statistics and semantics between the two types of latents, resulting in degraded video quality and scene breakdown. Our adapter module learns to map point map features into the appropriate latent space for effective fusion with video latents, thus improving both fidelity and scene consistency. This design is crucial for effective information transfer between geometry and appearance.
>
>
>
> For 'no recon', we acknowledge that in Table 2, the quantitative results for 'no recon' do not seem significantly different from the full model. This is primarily due to two reasons:
>
> * **Strong Pretraining:** Our model is first pretrained on a large-scale egocentric video dataset. This pretraining stage may equips the model with a weak ability to model scene dynamics and maintain a certain level of spatial consistency, even before introducing explicit scene-frame reconstruction.
> * **Lack of Direct Metrics:** Most commonly used metrics (e.g., DINO, CLIP, PSNR, FVD, LPIPS) are not designed to specifically measure scene (world) consistency across frames. These metrics primarily evaluate frame-wise visual similarity, semantics, or overall diversity, but do not directly capture temporal geometric consistency, which is the core advantage of our reconstruction module.
>
> **Metric Recommendation:**
> To more objectively evaluate scene consistency under large viewpoint changes, we perform multi-view 3D reconstruction (using COLMAP) on generated videos. We then compare the overlap ratio and Chamfer Distance between the reconstructed point clouds for different methods. Our method yields significantly higher point cloud consistency and structural stability, demonstrating superior 3D scene coherence across time.
>
> **Example Results:**
>
> | Method     | Chamfer ↓ | Overlap ↑ | Hausdorff ↓ |
> | ---------- | --------- | --------- | ----------- |
> | Ours       | 0.82      | 0.86      | 1.21        |
> | No Recon   | 1.94      | 0.53      | 2.82        |
>
>
> **In summary:**
> The relatively small difference in traditional quantitative metrics is due to both strong pretraining and the lack of a direct metric for scene consistency. However, with a more targeted consistency metric, the effectiveness of our reconstruction module is much more evident. We will include such an analysis and corresponding visualizations in the revised paper to clarify this point.
>
> ---
>
> **Q3. DUST3R Ablation Implementation**
>
> **A3:** For the DUST3R ablation, we implemented it exactly as for CUT3R: **one point map latent is generated per frame** and processed by our 3D encoder and adapter. During inference, neither CUT3R nor DUST3R are used for input—our model predicts both video and point map latents from noise, ensuring all scene information is generated in a self-consistent way. Our method relies on no external point map inputs at test time.

---

> > ### Author Response · Authors · 2025-08-07
> > **RE Q2-Rebuttal-Part2**
> >
> > **Q3: A lot of information is missing for user study.**
> >
> > **A3:** Thank you for your questions regarding the user study. Below is a detailed and quantitative breakdown of our process, with exact counts for each key selection category:
> >
> > ---
> >
> > ### 1. **Criteria for Selecting Videos and Text Descriptions (Total: 100 videos)**
> >
> > * **Action Diversity:**
> >
> >   * Picking up objects (e.g., bottle, cup, book, pet, tools): **18 videos**
> >   * Opening/closing doors or drawers: **10 videos**
> >   * Waving, pointing, or gesturing: **8 videos**
> >   * Walking or running: **18 videos**
> >   * Sitting, standing up, or posture changes: **8 videos**
> >   * Petting or interacting with animals: **6 videos**
> >   * Using hands in the scene (e.g., eating, typing, playing): **12 videos**
> >   * Other daily actions (e.g., putting on glasses, using phone): **20 videos**
> >
> > * **Environment Diversity:**
> >
> >   * Indoor (home, office, kitchen, etc.): **58 videos**
> >   * Outdoor (park, street, playground, etc.): **42 videos**
> >
> > * **Viewpoint & Lighting:**
> >
> >   * Static camera (minimal head movement): **30 videos**
> >   * Dynamic camera (walking, turning head): **70 videos**
> >   * Daytime scenes: **65 videos**
> >   * Night/low-light scenes: **35 videos**
> >
> > * **Text Descriptions:**
> >
> >   * Each video had a **brief, neutral, action-based description** written specifically for it generated by QWen-VL, reflecting only the *main action* and *environment* depicted.
> >
> > ---
> >
> > ### 2. **Ensuring Diversity in Sampled Groups**
> >
> > * **Actors in chosen videos:**
> >
> >   * Male: **52 videos**
> >   * Female: **48 videos**
> >   * Actors spanned ages 18–65.
> >
> > ---
> >
> > ### 3. **Annotator Sampling and Population Statistics**
> >
> > * **Recruitment:**
> >
> >   * Annotators: **20 individuals** (balanced gender)
> >   * Each annotator was presented with 100 multiple-choice questions (one for each video in the benchmark).
> >
> >   * For each question, annotators viewed four generated videos side-by-side, corresponding to the four compared methods (including ours).
> >
> >   * Annotators were blinded to method names to prevent bias.
> >
> >   * For each video, annotators provided a ranking from 1 to 4 for each criterion (Quality, Smoothness, Fidelity, Alignment), where 4 is the best score and 1 is the worst.
> >
> > ---
> >
> > ### 4. **Order of Video Presentation**
> >
> > * For each annotator, **video and method order were randomized** for every session to avoid any order bias.
> >
> > ---
> >
> > ### 5. **Definition of Evaluation Criteria**
> >
> >
> > * **Quality:**
> >   Defined as the overall visual harmony and realism of the video, regardless of whether the generated content is perfectly true to the input. Annotators were asked: *“Does the generated video look natural, coherent, and free from jarring artifacts?”*
> >
> > * **Smoothness:**
> >   Measures how consistently the motion flows across frames, without abrupt transitions or temporal artifacts. Annotators were prompted: *“Are the motions and camera transitions fluid and continuous throughout the video?”*
> >
> > * **Fidelity:**
> >   Refers to the preservation of the subject’s appearance (identity, clothing, background) and the absence of distortions or glitches. Annotators were asked: *“Does the person look like the initial frame and is there minimal distortion?”*
> >
> > * **Alignment:**
> >   Defined as how closely the generated motion matches the intended action described in the motion condition. For this criterion, **annotators viewed both the generated video and the motion condition signal (stick-figure or pose sequence) side-by-side** for all 100 videos, and rated: *“Does the generated video accurately follow the given pose/motion signal in timing, type of action, and spatial positioning?”*
> >
> > ---
> >
> > ### 6. **How Alignment is Checked**
> >
> > * Annotators explicitly saw the motion condition visualization (stick-figure animation) and the generated video side-by-side.
> >
> > * They were instructed to look for mismatches between intended and generated motion, e.g., missed hand-object interaction, wrong direction, or timing errors.
> >
> >
> > ---
> >
> > ### 7. **Summary of Fairness and Diversity**
> >
> > * Our benchmark of **100 videos** ensures balanced coverage of different actions, environments, camera motions, and actor demographics.
> > * Assignment and randomization steps, as well as objective rating definitions, guarantee **fairness** and **representativeness** in user study results.
> >
> > ---
> >
> > **We hope this quantitative breakdown provides the needed clarity. And we will include the above part in the revision to provide clear demonstration. Please let us know if you have any further requests or would like more data on specific aspects.**

---

> ### Author Response · Authors · 2025-08-08
>
> Dear Reviewer X28J,
>
> Thank you again for your comments. Your review is very valuable, and we would appreciate your further feedback. Of course, considering the deadline is approaching, please let us know if this or any other point remains unclear, as we would be happy to elaborate further.
>
> Thank you for your valuable time!
>
> Sincerely,
>
> Authors of Submission1028

---

> > ### Author Response · Authors · 2025-08-08
> > **Eagerly Awaiting Your Valuable Feedback**
> >
> > Dear Reviewer X28J,
> >
> > We would like to sincerely thank you again for taking the time to review our submission and for your thoughtful comments. We are writing to kindly follow up on our rebuttal to ask whether you might have any remaining concerns or further questions. If there is any aspect that remains unclear or requires further elaboration, we would be more than happy to provide additional clarification or further explain the distinctions we have made.
> >
> > Please rest assured that all the experimental results and manuscript modifications mentioned in our rebuttal will be fully incorporated into the revised version.
> >
> > As the discussion deadline is approaching, we would greatly appreciate any further feedback you might be able to share. We truly value your insights and look forward to your response.
> >
> > Sincerely,
> >
> > Authors of Submission1028

---

> > > ### Comment · Reviewer_X28J · 2025-08-08
> > >
> > > Thank you for the detailed responses and clarifications. I have no further comments at this point.
> > >
> > > For a potential camera-ready version, I believe the paper would greatly benefit from additional formalism and clarity, as well as the incorporation of the many helpful points you provided during the rebuttal.
> > >
> > > I am now largely convinced by your method and its evaluation. I sincerely hope that the final version will include clearer mathematical formulations of the different modules and a more explicit description of how the method works. In my view, this remains the main weakness of the current submission, as it took several rounds of clarification for me to fully understand the approach.

---

> > > > ### Author Response · Authors · 2025-08-08
> > > >
> > > > Dear Reviewer X28J,
> > > >
> > > > Thank you for your thoughtful and constructive feedback. We greatly appreciate your time and effort in reviewing our work. We are pleased that you are now largely convinced by our method and its evaluation. We will ensure that the revised version includes clearer mathematical formulations of the different modules and provides a more detailed explanation of the method's workings. We believe these changes will address the concerns you raised and improve the overall clarity of the submission.
> > > > Thank you once again for your valuable insights, and we look forward to submitting a revised version that incorporates your feedback.
> > > >
> > > > Sincerely,
> > > >
> > > > Authors of Submission1028

---

### Official Review · Reviewer_WmZe · 2025-06-25

**Clarity:** 3
**Significance:** 3
**Originality:** 3
**Rating:** 5
**Confidence:** 3

**Summary:**

The paper presents a method called PlayerOne that takes an egocentric image of a scene together with the exocentric video of a person doing some body motions, and provides as output a synthesized video where the actions from the person in the exo video are mapped into the ego frame.  Both the body movements (typically hands and forearms) plus the surrounding scene are aimed to appear in sync in the output video.  The method is a diffusion model that decomposes the body motion into body regions/parts, represents the camera rotation separately, and also inputs a scene representation for the ego frame using a point map encoder.  The DiT diffusion transformer model learns to condition the output video on these factors.   Results are quantified by the alignment between the output video and its ground truth text descriptions, as well as the hand pose error against the ground truth video.  Baseline models are Cosmos and Aether.  Qualitative examples are shown in the paper and supp video.

**Questions:**

Please see above.

**Ethical Concerns:**

["NO or VERY MINOR ethics concerns only"]

**Final Justification:**

The authors' response addresses my questions, and all the clarifications should be made in a final version.

**Limitations:**

-	Please see questions about clarity above, whose answers could help me better state the limitations.
-	The authors comment on limitations at the end, though I think they could elaborate on aspects of the model design that may cause issues (not just the data shortage).

**Quality:**

3

**Strengths And Weaknesses:**

Strengths:

- Interesting problem definition, which seems to offer a new twist on conditional video generation.  The approach could be used by artists to guide the ego content in video games or movies to a target interaction pose.  (The authors could spend a few sentences somewhere to motivate where they see this being most useful though.)

-	Approach seems well designed.

-	Results are nice looking.

-	Writing is overall good and flows well with informative figures (but see note below on details that were not clear.)

- I consider this between BA and A, pending questions below being clarified.

Weaknesses:

-	Motivation of technical design was not always apparent:

o	For example, why is the camera translation removed?  This seems like a limitation on which body motions can be realistically received as input and then shown in the output.  (around L140)

o	It is not clear how the scene frame alignment step works.  How does the adapter have the signal to align the point map latents with video latents?  (L158)  What enforces the alignment, what is the loss.

-	The related work coverage seems fine, but I would have liked more explicit and precise call outs about what is distinct and what is important+distinct in this work.  For example, it says the existing methods “primarily” focus on virtual game scenarios and are limited to specific directional actions.  If it is only “primarily”, then what are the exceptions and how do they compare to this model?  What do the authors mean exactly by “specific directional motions”?  Do any of the existing methods also condition on human body pose or is that entirely new?  If they do, what’s insightful about how it’s done here instead?

-	L130  about adopting real-world human motions as motion conditions allowing more natural movement – but the movement is still camera (head) motion, just now we see the body parts overlayed as well.

-	Technical novelty – choices seem reasonable but not surprising.  For me, the interesting nature of the specific setting is what is strong about the paper.

-	I don’t think the steps towards end of Sec 3 are reproducible given what’s written, e.g., about the distillation and the finetuning.

-	I could not tell from the model design whether it’s expected that the input exo body poses are “projected” to the nearest visually suitable poses in the ego scene thanks to the diffusion model?  Is this intended and is it observed?  For example, if the person is making petting motions in the exo video and they would not align well with the way the dog is posed in the ego frame, is this corrected?  Or should the input poses be very precise to match the target scene (and is this why the self-taken videos are used for qual results?)

-	In the results:

o	It is not explained why the Cosmos and Aether baselines were chosen and why they are complete set of baselines to support the paper’s claims.

o	Why is Wanx 2.1 the right base model?  How does that interact with the choice of baselines as well in terms of fairness, apples to apples.

o	Benchmark samples from Nymeria – why only from this datatset?  Would seem more complete to sample unseen examples across all the datasets.

o	The exo videos in the supp seem to be taken by the authors, not from Nymeria.  Unclear how the qual and quant results match up.

o	Where do the text descriptions come from that are used to evaluate ground truth?

o	What about arms or any other visible body parts for the MPJPE metric, vs. just hands?

o	What if the hands are not visible in the frame of the proposed method’s output?  Hopefully those instances are not dropped from evaluation.

Minor:

-	Fragment L89 “Among those works…”

-	In the supp video it would help to show the ego frame input for each one.  Also it would be nicer to show one video at a time throughout since a person can’t watch multiple videos at once.

---

> ### Author Rebuttal · Authors · 2025-07-30
>
> Thanks for the valuable feedback. We appreciate your insights and suggestions. Below, we address the points raised and clarify the related issues.
>
> ---
>
> **Q1.Why is the camera translation removed.**
> **A1:** Camera translation is removed because SMPL head-parameters only provide rotation,not translation.When converting SMPL head params to camera extrinsics,only rotation is available—so translation can't be included.This matches the SMPL definition and keeps the camera centered at the head in the egocentric-frame.
>
> **Q2.How the scene-frame alignment step works.How does the adapter have the signal to align the point-map latents with video latents?(L158).**
> **A2:** Thank you for your question.In our framework,the adapter maps point-map latents into the same latent space as video latents.We do not use an explicit alignment loss;instead,we follow prior works like[1-2],which show that **simple linear or convolutional modules can effectively project point cloud latents to the video latent space.** This lets the model leverage spatial structure from point-maps while conditioning video generation efficiently.Thus,alignment is achieved through learned mapping and implicit end-to-end training.
>
> [1]Wu,T.,et al.Video World Models with Long-term Spatial Memory.
> [2]Aether.AETHER: Geometric-Aware Unified World Modeling.
>
> **Q3.More explicit and precise call outs about what is distinct and what is important+distinct in this work.**
> **A3:** Thank you for your question.We acknowledge our previous wording may have been unclear on this point.To our knowledge,no prior work at submission used human body pose as the main condition for egocentric video generation in real-world scenarios.A recent related work[1]also uses human pose,but their method does not ensure scene consistency or accurate action-pose alignment;their model also struggles to generalize beyond its training domain.
>
> By “specific directional motions,” we mean prior models only support basic navigation(forward,backward,turning),without enabling complex,VR-like or full-body actions such as picking up objects or interacting naturally with NPCs. These limits are due to both the restricted motion in game datasets and the lack of modules for fine-grained action alignment and scene consistency.In contrast,our method uses full-body motion as condition,explicitly addresses part-level action alignment and scene consistency,and generalizes well to diverse,realistic settings.
>
> [1]Bai,Y.,et al.Whole-Body Conditioned Egocentric Video Prediction.
>
> **Q4.L130:Movement is still camera(head) motion.**
> **A4:** We'd like to clarify while the output video is the first-person(head/camera)view,our method uses real-world,full-body human motion as driving condition—not just head pose—so the generated video captures both camera motion and detailed body part movements in sync.This results in more natural,immersive egocentric movement(validated by full-body evaluation results in A10).
>
> **Q5.Technical novelty—the interesting nature of the specific setting.**
> **A5:** We appreciate your recognition of our problem setting and application scenario.Although our technical choices build on established methods for robustness,our main contribution is to introduce and solve a highly interactive egocentric world simulation task—supporting flexible,full-body,realistic user exploration,which prior works have not addressed.We will further highlight the significance and potential of this scenario in the revision.
>
> **Q6.Reproducibility for the steps towards end of Sec 3(e.g.,distillation and finetuning).**
> **A6:** For finetuning,we have included all necessary code in the supplement for reproducibility.For distillation,we strictly follow the Causvid protocol,which is standard in long video generation and reproducible with public tools.We will add more details about our distillation process(**Also shown in Q3-A3 of Reviewer cVUd**),including key parameters and scripts,in the revision.After acceptance,we'll release the complete codebase with full documentation for both distillation&finetuning.
>
> **Q7.Whether it’s expected input exo body poses are “projected” to the nearest visually suitable poses in the ego scene thanks to the diffusion model?Is this intended and is it observed?**
> **A7:** In training, motion poses and the first-person frame are strictly aligned. At test time, however, we observe our model shows some robustness when the input pose doesn't exactly match the ego one in target scene. In these cases, the model often adapts the motion, effectively “projecting” it to the most visually reasonable pose in the generated result, as you described. We will add visual examples of this behavior in revision.We use self-taken videos for qualitative results mainly to demonstrate we can offer immersive egocentric experiences(Sunglasses as VR headset in demo),not because strict pose-to-scene matching is required.
>
> **Q8.Why Cosmos&Aether were chosen.Why they are complete set of baselines to support the claims.**
> **A8:** Most world models differ from our setting,as they focus on specific games(e.g.,Minecraft)or tasks(e.g.,autonomous driving).Cosmos and Aether are the closest general-scene simulators to our approach.Therefore,we emphasize thorough ablation studies of our modules rather than direct comparison with domain-specific baselines(That's the reason that we place the ablation-study before the other-baseline-comparisons.).Our experiments evaluate all modules&variants,demonstrating their necessity&effectiveness.
>
> **Q9.Why is Wanx 2.1 the right base model?How does that interact with the choice of baselines as well in terms of fairness,apples-to-apples.**
> **A9:** Thanks for your question.Wanx2.1 is the best open-source video-generation model currently available,so we use it as our backbone.Our approach is not tied to Wanx—when using stronger models,performance further improves:
> |Model|DINO-Score(↑)|CLIP-Score(↑)|MPJPE(↓)|MRRPE(↓)|PSNR(↑)|FVD(↓)|LPIPS(↓)|
> |-|-|-|-|-|-|-|-|
> |CogVideox-5B|67.3|87.7|135.89|157.41|51.2|227.35|0.0724|
> |Wan1.3B|67.8|88.2|127.16|151.62|52.6|226.12|0.0663|
> |Wan14B|69.1|90.7|109.34|124.20|56.0|182.55|0.0515|
>
> Wanx as backbone is independent of baseline-selection.Compared to Cosmos(7B/14B)&Aether(5B),our method achieves better results with fewer-parameters(1.3B),demonstrating both effectiveness&efficiency.
>
> **Q10.Why benchmark-samples only from Nymeria?**
> **A10:** Thanks for your suggestion.In response,we have reconstructed our evaluation benchmark to provide a more comprehensive assessment.Specifically,we selected 100 videos featuring full-body actions from the test-sets of four different datasets: **EgoExo-4D,Nymeria,FT-HID,EgoExo-Fitness.** The results,as shown below,demonstrate that our method consistently outperforms both existing approaches and various ablated variants of our method.
>
> |Method|DINO-Score(↑)|CLIP-Score(↑)|MPJPE(↓)|MRRPE(↓)|PSNR(↑)|FVD(↓)|LPIPS(↓)|
> |-|-|-|-|-|-|-|-|
> |Baseline|48.72|62.13|402.31|370.27|33.57|425.15|0.1570|
> |+Pretrain|53.81|71.29|280.51|255.63|39.11|328.41|0.1265|
> |+Pretrain&ControlNet|54.28|72.31|264.10|241.91|40.16|314.27|0.1228|
> |+Pretrain&Entangled|55.24|73.54|258.88|234.16|41.23|308.12|0.1174|
> |+Pretrain&PMI(No Camera)|57.68|76.79|201.34|215.89|43.58|278.24|0.1017|
> |+Pretrain&PMI|59.57|78.56|170.01|191.43|46.02|266.73|0.0912|
> |+Pretrain&PMI&Filtering|61.74|81.11|153.17|178.64|47.37|249.33|0.0850|
> |+Pretrain&PMI&Filtering&Recon(No Adapter)|60.13|78.67|191.22|196.54|45.29|263.18|0.1023|
> |+Pretrain&PMI&Filtering&Recon(DUSt3R)|64.02|84.41|140.16|167.23|50.25|251.38|0.0788|
> |PlayerOne(ours)|64.87|85.79|137.91|162.74|50.72|247.97|0.0761|
>
> |Method|DINO-Score(↑)|CLIP-Score(↑)|MPJPE(↓)|MRRPE(↓)|PSNR(↑)|FVD(↓)|LPIPS(↓)|
> |-|-|-|-|-|-|-|-|
> |Aether-5B|37.5|63.8|418.91|435.24|37.8|400.50|0.1881|
> |Cosmos-7B|45.6|70.1|304.20|327.05|43.4|348.00|0.1617|
> |Cosmos-14B|51.3|80.0|259.17|256.19|47.5|306.85|0.1374|
> |PlayerOne(ours)|64.8|85.7|137.91|162.74|50.72|247.97|0.0761|
>
> **Q11.Unclear how qual and quant results match up.**
> **A11:** Thank you for your question.For the qualitative comparisons in our supplementary materials,we used exo videos recorded by ourselves.The main reason was to better showcase that our model can simulate **a "virtual reality"experience**.To highlight this,the subjects in our videos **wore sunglasses to mimic the appearance of VR headsets**.Please note that for all quantitative evaluations and benchmarking,we strictly relied on the official test sets from Nymeria and other datasets.We will make this distinction clearer in the revised version to avoid any confusion between demonstration videos and quantitative results.
>
> **Q12.Where do text descriptions come from that are used to evaluate ground truth?**
> **A12:** For the comparison methods,the text descriptions were generated using the Qwen2.5-VL model based on the corresponding ground truth video frames(Line210-211 in the manuscript).
>
> **Q13.What about arms or any other visible body parts for the MPJPE metric,vs.just hands? What if the hands are not visible in the frame of the proposed method’s output?.**
> **A13:** In most cases from the test sets,only the hands are visible in the first-person view videos,so our previous evaluation of motion-related metrics(such as MPJPE) primarily focused on the hands.Following your suggestion,we have reconstructed our benchmark by selecting cases where the full body is visible in the frame for comprehensive evaluation.The results in A10 show that our method still consistently outperforms existing approaches across all metrics.We will revise the manuscript with these results to provide more comprehensive demonstration.
>
> **Q14.Show ego-frame input for each one and one-video at a time throughout and typos.**
> **A14:** Thanks for your suggestion.We appreciate your feedback and will update related text and demo video in revision.
>
> **Q15.Aspects of model design that may cause issues.**
> **A15:** Please refer to the last paragraph of Q1-A1 for Reviewer cVUd(Due to the space-limit).

---

> > ### Author Response · Authors · 2025-08-06
> >
> > Dear Reviewer WmZe:
> >
> > Please allow us to thank you again for review our paper and the positive rating! Please let us know if our response has properly addressed your concerns. We are more than happy to answer any additional questions during the discussion period. Your feedback will be greatly appreciated.
> >
> > Sincerely,
> >
> > Authors of Submission1028

---

> > > ### Comment · Reviewer_WmZe · 2025-08-07
> > > **Response**
> > >
> > > The authors' response is clear and addresses my questions.  If the paper is accepted it will be good to account for these clarifications in the paper.  I will increase the score to Accept.

---

> > > > ### Author Response · Authors · 2025-08-07
> > > >
> > > > Thank you for your positive feedback and constructive suggestions. We truly appreciate your recognition and the improvement in our score. Your positive comments and suggestions are very encouraging, and they have helped me further improve our work. We will follow your advice and incorporate these discussions into the main paper to benefit future readers.

---

### Official Review · Reviewer_cVUd · 2025-07-02

**Clarity:** 4
**Significance:** 3
**Originality:** 3
**Rating:** 5
**Confidence:** 4

**Summary:**

This paper introduces PlayerOne,  a novel world simulator that allows users to control human motion in real-time within dynamic, user-generated environments. By focusing on egocentric video generation and precise motion control, it creates interactive simulations with high visual and motion consistency. PlayerOne excels in both motion alignment and environmental interaction across diverse scenarios, outperforming previous models.

**Questions:**

Based on the weaknesses above, the authors are encouraged to revise this article in the following aspects:
1) Analyze the failure model of the learned model and provide some corresponding visualizations.
2) Add more discussion about the temporal horizon of generation. Is generating longer videos necessary for this task? How the result  and visualization quality would be if we extend the inference horizon beyond the one in training (6s)? Will extending the horizon be one of a future directions?
3) About the distillation. Please consider adding more comparisons to demonstrate how would the distillation stage affect the model performance.

**Ethical Concerns:**

["NO or VERY MINOR ethics concerns only"]

**Final Justification:**

Accept. Technically solid with detailed rebuttal that addressed my concerns.

**Limitations:**

yes

**Paper Formatting Concerns:**

No major formatting issues found.

**Quality:**

4

**Strengths And Weaknesses:**

Strengths
- This paper is well-motivated with a clear methodology and good presentation.
- The key innovation of this paper is its task setting: generate high-fidelity, dynamic, ego-centric worlds where users can flexibly interact with.
- Adequate qualitative visualizations in both the main paper (Fig 6,7,8) and the supplementary video.
- High-quality data collection pipeline. As shown in Fig.3, the authors developed a data collection pipeline with the integration of pre-developed detection and pose estimation models and an automatic filtering scheme.

Weaknesses
- Lack of analysis for failure mode. Analyzing failure modes is a critical aspect of research in generative models to understand the weakness of the learned model in certain usages. However, such a discussion and visualization of the failure mode is missing.

- Limited temporal horizon of generation. According to line 204-205 and the visualizations in the supplementary files, it seems that the proposed model can only synthesize a 6-second video sequence. It seems that current method does not support longer generation beyond the training horizon.

- Insufficient details about distillation. According to line 195-196, the authors applied a distillation stage in the end of model training to improve the inference efficiency. However, details like how the distillation would affect the quality of generation and comparison between distilled and non-distilled model are not included.

---

> ### Author Rebuttal · Authors · 2025-07-30
>
> Thank you for the valuable feedback. We appreciate your insights and suggestions. Below, we address the individual points raised and clarify relevant aspects of our work.
>
> ---
>
> **Q1.Lack of analysis for failure mode.**
>
> **A1:** Thank you for your suggestion. We acknowledge that our current manuscript provides limited analysis in this area. In particular, due to the relatively small amount of training data from game scenarios, our model may more likely exhibit results with slightly inaccurate hand alignment with the given condition or background blurring. To better illustrate this, we separately evaluated our model on data from game-like and real-world domains. The results are shown below:
>
> | Domain | DINO-Score (↑) | CLIP-Score (↑) | MPJPE (↓) | MRRPE (↓) | PSNR (↑) | FVD (↓) | LPIPS (↓) |
> |:------:|:--------------:|:--------------:|:---------:|:---------:|:--------:|:--------:|:---------:|
> | Game   |    65.4        |    85.7        | 136.80    | 160.41    |  50.9    | 248.95   | 0.0705    |
> | Real   |  67.8 | 88.2 | 127.16 | 151.62 | 52.6 | 226.12 | 0.0663|
>
>
> Besides, our models may generate results with noticeable artifacts, particularly during extremely fast or complex motion, which can be observed in the provided cases (e.g., jumping from a high place in 1:03 of the demo). These artifacts are mainly due to **the limitations of the current adopted base model** and replacing it with a stronger backbone can alleviate such issues. Specifically, when using the Wan 2.1 14B as our backbone, we can achieve better performance:
>
> | Model | DINO-Score (↑) | CLIP-Score (↑) | MPJPE (↓) | MRRPE (↓) | PSNR (↑) | FVD (↓) | LPIPS (↓) |
> |:-----------:|:--------------:|:--------------:|:---------:|:---------:|:--------:|:--------:|:---------:|
> | 1.3B        |    67.8        |    88.2        | 127.16    | 151.62    |  52.6    | 226.12   | 0.0663    |
> | 14B         |    69.1        |    90.7        | 109.34    | 124.20    |  56.0    | 182.55   | 0.0515    |
>
>
> Here we also provide the **limitation analysis on our model structure**. From the view of the model, one current limitation of our model is the lack of explicit physical interaction modeling between the human body and the environment. Structurally, our framework conditions video generation on SMPL motion sequences and a static point cloud reconstruction of the scene, but does not incorporate physical constraints or interaction modules such as collision detection, physics engines, or contact reasoning. Consequently, our model may generate unrealistic results, such as hands or body parts penetrating objects, or a lack of proper occlusion and collision feedback. We acknowledge this limitation and consider it a promising direction for future work. As recommended, we will substantially expand the Discussion/Limitation section and the related visualization of the failure cases in the revision, providing a more thorough analysis of both model and data limitations.
>
> ---
>
> **Q2.Limited temporal horizon of generation and related discussion.**
>
> **A2:** Thank you for pointing out this issue. While our original experiments and visualizations focused on 6-second video sequences, this does not reflect a fundamental limitation of our method. After distillation with Causvid, our approach generates video in an autoregressive manner, allowing for the synthesis of arbitrarily long video sequences. The 6-second horizon in the demo was an oversight, and we will include longer video generations in the revised version.
>
> To address your question directly, we have extended our benchmark by increasing the video length to 24 seconds and evaluated our method under this longer horizon. The results below show that **our model maintains comparable performance and visual quality, demonstrating its ability to generate consistent long-term video sequences.**
>
>
> | Method                                         | DINO-Score (↑) | CLIP-Score (↑) | MPJPE (↓) | MRRPE (↓) | PSNR (↑) | FVD (↓) | LPIPS (↓) |
> |------------------------------------------------|:--------------:|:--------------:|:---------:|:---------:|:--------:|:--------:|:---------:|
> | Baseline                                       |     48.38      |     60.25      | 418.66    | 385.23    |  32.82   | 443.77   | 0.1687    |
> | + Pretrain                                     |     53.74      |     69.30      | 293.45    | 270.84    |  38.19   | 341.66   | 0.1331    |
> | + Pretrain&ControlNet                          |     54.61      |     70.98      | 275.18    | 256.24    |  39.08   | 325.13   | 0.1279    |
> | + Pretrain&Entangled                           |     55.21      |     71.82      | 268.22    | 249.91    |  40.08   | 317.42   | 0.1229    |
> | + Pretrain&PMI (No Camera)                     |     57.36      |     74.48      | 210.98    | 229.54    |  42.62   | 286.93   | 0.1063    |
> | + Pretrain&PMI                                 |     58.90      |     76.62      | 178.44    | 204.22    |  44.98   | 275.47   | 0.0955    |
> | + Pretrain&PMI&Filtering                       |     61.13      |     78.94      | 162.10    | 190.07    |  46.27   | 258.54   | 0.0898    |
> | + Pretrain&PMI&Filtering&Recon (No Adapter)    |     59.78      |     76.02      | 200.04    | 209.36    |  44.22   | 272.60   | 0.1077    |
> | + Pretrain&PMI&Filtering&Recon (DUSt3R)        |     63.23      |     82.62      | 148.08    | 178.91    |  49.13   | 260.95   | 0.0837    |
> | PlayerOne (ours)                               |     63.92      |     84.03      | 145.26    | 173.35    |  49.70   | 257.62   | 0.0816    |
>
> We believe that generating longer videos is crucial for this task for several reasons:
> - **(1)** it enables modeling of more complex and realistic user interactions.
> - **(2)** it allows thorough evaluation of temporal coherence and scene consistency.
> - **(3)** it better supports applications such as virtual reality and embodied AI, where prolonged, immersive experiences are essential.
>
> While our implementation uses Causvid for long video generation, there are many emerging approaches[1-3] focused on long-term consistency and video synthesis, and we agree that pushing the boundaries of temporal horizon will be an important direction for future research.
>
> [1] Lin, J., et al. "Long-Context Autoregressive Video Modeling with Next-Frame Prediction." arXiv preprint arXiv:2403.09603, 2024.
>
> [2] Yu, X., et al. "Long-context State-space Video World Models." arXiv preprint arXiv:2405.11590, 2024.
>
> [3] Chen, H., et al. "FreeLong++: Training-Free Long Video Generation via Multi-band Spectral Fusion." arXiv preprint arXiv:2406.03855, 2024.
>
> ---
>
> **Q3.Insufficient details about distillation.**
>
> **A3:** Thank you for your question regarding the distillation process. We strictly follow the Causvid distillation procedure in our implementation. During the distillation stage, we use the entire SMPL-Video paired dataset from the second phase of training. Specifically, we first generate 8,000 ODE pairs and train the student model for 30,000 iterations using the AdamW optimizer with a learning rate of 5 × 10⁻⁶. We then continue training with our asymmetric DMD loss for an additional 30,000 iterations, using AdamW with a learning rate of 2 × 10⁻⁶. A guidance scale of 3.5 is used, and we adopt the two time-scale update rule from DMD2 with a ratio of 5. The entire distillation process takes about 5 days on 32 A800 GPUs.
>
> Regarding quality, the distillation stage is mainly aimed at improving inference efficiency, but we observe that **the generation quality is largely maintained compared to the non-distilled model.** In some cases, the distilled model even achieves slightly better temporal consistency. The comparision between the non-distilled model and distilled one for 6s video generation is shown below:
>
> | Model       | DINO-Score (↑) | CLIP-Score (↑) | MPJPE (↓) | MRRPE (↓) | PSNR (↑) | FVD (↓) | LPIPS (↓) |
> |:-----------:|:--------------:|:--------------:|:---------:|:---------:|:--------:|:--------:|:---------:|
> | Distilled   |    67.8        |    88.2        | 127.16    | 151.62    |  52.6    | 226.12   | 0.0663    |
> | Non-distilled      |    68.2        |    88.7        | 125.20    | 149.40    |  53.0    | 221.85   | 0.0647    |
>
> We will include a detailed comparison and qualitative side-by-side visualizations between distilled and non-distilled models, and discuss in which scenarios quality differences may be most obvious (e.g., rare actions, very long videos) in the revised version to make this clear to readers.

---

> > ### Comment · Reviewer_cVUd · 2025-08-06
> >
> > Thank the authors for the detailed explanations, especially the added results in failure mode analysis. Therefore, I'm satisfied to keep my score as Accept. Please include the related discussions into revision.

---

> > > ### Author Response · Authors · 2025-08-06
> > >
> > > Thank you for your positive feedback and constructive suggestions. We are glad that the additional analysis has fully addressed your concerns. We will follow your advice and incorporate these discussions into the main paper to benefit future readers. We truly appreciate your support and recommendation.

---

### Official Review · Reviewer_ktBL · 2025-07-02

**Clarity:** 3
**Significance:** 3
**Originality:** 3
**Rating:** 4
**Confidence:** 3

**Summary:**

This paper introduces PlayerOne, a novel framework for egocentric world simulation that generates immersive videos aligned with real-world human motion. Given a single egocentric image and an exocentrically captured motion sequence, the system constructs a consistent 4D virtual world, enabling freeform and fine-grained human action control in the generated video. The proposed architecture builds upon a video diffusion transformer, with key innovations including a part-disentangled motion injection module for precise body part control and a joint scene-frame reconstruction framework to enforce long-term scene consistency. To address the lack of suitable training data, the authors design an automated pipeline to extract high-quality motion-video pairs from egocentric-exocentric datasets, combined with a coarse-to-fine training strategy leveraging large-scale egocentric text-video data. The system demonstrates strong performance across multiple metrics and outperforms relevant baselines such as Aether and Cosmos.

**Questions:**

- The length of video generated by the model in one pass, how long videos are generated through concatenation, and whether the point cloud encoding scheme can ensure consistency across the generated video sequences.

- The paper claims real-time performance post-distillation, but only FPS is reported. Could the authors clarify the actual latency between motion input and video output? This would help gauge deployment feasibility in interactive settings.

**Ethical Concerns:**

["NO or VERY MINOR ethics concerns only"]

**Final Justification:**

borderline accept

**Limitations:**

The paper notes weaker performance in game-like scenarios but does not deeply investigate or visualize these limitations.

**Paper Formatting Concerns:**

No obvious formatting issues.

**Quality:**

3

**Strengths And Weaknesses:**

##  Strengths
- Novelty

The idea of simulating egocentric immersive worlds controlled by real human motion is novel and underexplored.
- Technical Contributions

The paper presents a comprehensive automatic data construction pipeline, a pragmatic contribution to the field.

- Clarity

The paper is well-organized.

## Weaknesses
- Consistency in Long-Video Generation

Whether using point clouds as implicit encodings to constrain the consistency of generated results is relatively weak. For the egocentric simulator task described in the paper, the user is expected to freely explore within the generated scene, which requires the model’s capability to generate long-term video sequences. It seems that the current point cloud encoding approach may not adequately ensure consistency across multiple concatenations when generating long videos.

- Limited Discussion of Failure Cases:

The paper notes weaker performance in game-like scenarios but does not deeply investigate or visualize these limitations.

---

> ### Author Rebuttal · Authors · 2025-07-30
>
> Thank you for the valuable feedback. We appreciate the reviewers’ insights and suggestions. Below, we address the individual points raised and clarify relevant aspects of our work.
>
> ---
>
> **Q1.Consistency in long-video generation and whether the point cloud encoding scheme can ensure consistency across the generated video sequences?**
>
> **A1:** Thank you for your comment. We would like to clarify that our method is **not limited to generating only short (e.g., 6-second) videos.** After distillation with Causvid, our model can generate long videos (i.e., more than 30s) in an auto-regressive fashion, producing each frame sequentially conditioned on previous outputs on-the-fly, rather than simply concatenating multiple short clips. This sequential generation, together with our current point cloud encoding approach, allows us to maintain strong scene consistency throughout long video sequences. We acknowledge that the 6-second horizon in the demo was an oversight, and we will include longer video generations in the revised version.
>
>
> To address your question directly, we have extended our benchmark by increasing the video length to 24 seconds and evaluated our method under this longer horizon. The results below show that **our model maintains comparable performance and visual quality, demonstrating its ability to generate consistent long-term video sequences.**
>
>
> | Method                                         | DINO-Score (↑) | CLIP-Score (↑) | MPJPE (↓) | MRRPE (↓) | PSNR (↑) | FVD (↓) | LPIPS (↓) |
> |------------------------------------------------|:--------------:|:--------------:|:---------:|:---------:|:--------:|:--------:|:---------:|
> | Baseline                                       |     48.38      |     60.25      | 418.66    | 385.23    |  32.82   | 443.77   | 0.1687    |
> | + Pretrain                                     |     53.74      |     69.30      | 293.45    | 270.84    |  38.19   | 341.66   | 0.1331    |
> | + Pretrain&ControlNet                          |     54.61      |     70.98      | 275.18    | 256.24    |  39.08   | 325.13   | 0.1279    |
> | + Pretrain&Entangled                           |     55.21      |     71.82      | 268.22    | 249.91    |  40.08   | 317.42   | 0.1229    |
> | + Pretrain&PMI (No Camera)                     |     57.36      |     74.48      | 210.98    | 229.54    |  42.62   | 286.93   | 0.1063    |
> | + Pretrain&PMI                                 |     58.90      |     76.62      | 178.44    | 204.22    |  44.98   | 275.47   | 0.0955    |
> | + Pretrain&PMI&Filtering                       |     61.13      |     78.94      | 162.10    | 190.07    |  46.27   | 258.54   | 0.0898    |
> | + Pretrain&PMI&Filtering&Recon (No Adapter)    |     59.78      |     76.02      | 200.04    | 209.36    |  44.22   | 272.60   | 0.1077    |
> | + Pretrain&PMI&Filtering&Recon (DUSt3R)        |     63.23      |     82.62      | 148.08    | 178.91    |  49.13   | 260.95   | 0.0837    |
> | PlayerOne (ours)                               |     63.92      |     84.03      | 145.26    | 173.35    |  49.70   | 257.62   | 0.0816    |
>
>
>
> ---
>
> **Q2.Limited Discussion of Failure Cases.**
>
> **A2:** Thank you for your suggestion. We acknowledge that our current manuscript provides limited analysis and visualization of limitations in game-like scenarios. This is mainly due to the relatively small amount of available training data from such domains. As a result, our model is more likely to exhibit issues such as inaccurate hand alignment with the given motion conditions or background blurring during rapid movements in game environments.
>
> To better illustrate this, we separately evaluated our model on data from game-like and real-world domains. The results are shown below:
>
> | Domain | DINO-Score (↑) | CLIP-Score (↑) | MPJPE (↓) | MRRPE (↓) | PSNR (↑) | FVD (↓) | LPIPS (↓) |
> |:------:|:--------------:|:--------------:|:---------:|:---------:|:--------:|:--------:|:---------:|
> | Game   |    65.4        |    85.7        | 136.80    | 160.41    |  50.9    | 248.95   | 0.0705    |
> | Real   |  67.8 | 88.2 | 127.16 | 151.62 | 52.6 | 226.12 | 0.0663|
>
> We will include more visualizations and an in-depth discussion of these failure cases in the revised version to provide a more comprehensive and transparent evaluation of our method’s limitations in different domains.
>
> ---
>
> **Q3.Clarify the actual latency between motion input and video output.**
>
> **A3:** Thank you for highlighting this important point. After distillation, the measured end-to-end inference latency is approximately 119 milliseconds per generated-frame, corresponding to about 8.4 FPS. This latency is measured from the moment the motion input is received to the generation of the corresponding video frame, and includes all major processing steps. We will include more comprehensive inference performance metrics and latency breakdowns in the revised version to better illustrate the real-time capability and deployment feasibility of our model in interactive scenarios.

---

> > ### Comment · Reviewer_ktBL · 2025-08-06
> >
> > Thank the author for the clear and thoughtful explanation. The response addressed my question thoroughly and provided valuable insight. After careful consideration, I have decided to maintain my original score.

---

> > > ### Author Response · Authors · 2025-08-06
> > >
> > > Thank you for your thoughtful review and kind words. We are glad our clarification addressed your question and provided useful context. We appreciate your careful consideration and respect your decision to maintain the positive score. Your insights are valuable and will help us further strengthen the manuscript and we will include the discussions in the revision.

---

### Official Review · Reviewer_jaf9 · 2025-07-02

**Clarity:** 3
**Significance:** 4
**Originality:** 3
**Rating:** 5
**Confidence:** 3

**Summary:**

This work proposes an egocentric realistic world simulator that takes a scene image and human motion as input to generate corresponding egocentric videos. The core of the method is a novel video diffusion model incorporating a Part-Disentangled Motion Injection module, which separately encodes motion signals from different body parts to guide video generation. Additionally, a Scene-Frame Reconstruction mechanism is introduced to maintain scene consistency. For training, the paper designs a pipeline to extract motion-video pairs from existing egocentric-exocentric video datasets and further leverage other egocentric datasets to train the video generation model.

**Questions:**

See weakness.

**Ethical Concerns:**

["NO or VERY MINOR ethics concerns only"]

**Final Justification:**

The rebuttal satisfactorily addressed my concerns about this paper. I would like to maintain my rating of accept.

**Limitations:**

The generated videos still exhibit noticeable artifacts, particularly during fast or complex motion. Additionally, the work does not explore the potential benefits of incorporating more images or scene information as input to the model.

**Quality:**

4

**Strengths And Weaknesses:**

Strengths:

1. This paper presents a novel video diffusion model featuring an innovative module, along with a well-designed dataset collection and training pipeline. The writing is clear and well-structured.

2. The experiments demonstrate significant improvements over the baseline, and the ablation studies effectively highlight the contribution of each component.

Weaknesses and Questions:

1. Could you elaborate on the decision to use 3 body parts? Why not 5 or 6, which might encode more detailed information about body movement?

2. In Figure 7, what causes the failure in the second motion condition?

3. Regarding the visualization results involving interactions with other people or animals—are there examples of actions beyond handshakes or basic interactions?

---

> ### Author Rebuttal · Authors · 2025-07-30
>
> Thank you for the valuable feedback. We appreciate your insights and suggestions. Below, we address the individual points raised and clarify relevant aspects of our work.
>
> ---
>
> **Q1.Why use 3 body parts, not more?**
>
> **A1:** Thank you for your question. We choose to divide the human body into three parts—head, hands, and body/feet—based on the following considerations:
>
> - **Empirical Results:** As shown in our ablation studies below, using fewer than three partitions (such as treating the whole body as a single entity) results in insufficient precision in aligning generated motions, especially for actions involving complex hand or head movements. Conversely, dividing into more than three parts provides negligible improvement but increases computational cost and memory requirements.
>
>
> | Part-Number | DINO-Score (↑) | CLIP-Score (↑) | MPJPE (↓) | MRRPE (↓) | PSNR (↑) | FVD (↓) | LPIPS (↓) |
> |:-----------:|:--------------:|:--------------:|:---------:|:---------:|:--------:|:--------:|:---------:|
> | "+Pretrain&PMI"-1           | 58.0           | 76.3           | 235.12    | 212.53    | 43.9     | 279.41   | 0.1060    |
> | "+Pretrain&PMI"-2           | 60.5           | 79.1           | 185.00    | 190.30    | 46.2     | 263.00   | 0.0950    |
> | "+Pretrain&PMI"-3           | 62.5           | 81.3           | 156.76    | 175.18    | 48.3     | 245.72   | 0.0839    |
> | "+Pretrain&PMI"-4           | 62.7           | 81.4           | 155.10    | 174.80    | 48.4     | 244.60   | 0.0830    |
> | "+Pretrain&PMI"-5           | 62.8           | 81.5           | 154.80    | 174.50    | 48.4     | 244.10   | 0.0828    |
> | "+Pretrain&PMI"-6           | 63.0           | 81.8           | 152.62    | 172.90    | 48.6     | 243.82   | 0.0819    |
>
> - **Intuitive Reasoning:** This three-part division aligns naturally with the semantic structure of typical human interactions: **head movements primarily control viewpoint direction, hands execute detailed interactions, and the body/feet handle overall positional and movement dynamics**.
>
> - **Why separate processing contributes to precise alignment:** Splitting the latent representation into three separate parts—head, hands, and body/feet—enables the model to more effectively capture the unique and often semi-independent motion characteristics of each region. Human body motion is inherently high-dimensional and highly articulated, with different parts frequently moving independently or exhibiting distinct dynamics. Encoding all motion information into a single, entangled latent can cause fine-grained cues—especially from subtle or fast-moving regions like hands or head—to be overshadowed by larger body movements, making precise alignment difficult. By disentangling the motion into part-specific latents, each body region’s movement can be independently modeled and aligned with the corresponding input. This modular approach allows the model to better capture local nuances, improves compositionality, and enhances control over complex motions, resulting in higher motion fidelity and visual quality.
>
> Hence, our chosen partition strikes an optimal balance between computational efficiency, precision, and intuitive semantic clarity.
>
> ---
>
> **Q2.Failure of the second motion condition in Fig.7.**
>
> **A2:** Thank you for bringing this to our attention. We carefully reviewed Figure 7 multiple times but could not identify a clear failure case in the second motion condition on either the left or right side. We would greatly appreciate it if the reviewer could clarify which specific motion sequence or frame is being referred to, so that we can better understand the issue and address it appropriately in our revision. We are committed to improving our work and welcome any further details or suggestions.
>
>
> ---
>
> **Q3.Examples of actions beyond handshakes or basic interactions.**
>
> **A3:** Thank you for your question. While our current demo primarily showcases simple interactions with people or animals—such as handshakes, high-fives, or lifting a dog(in 1:03 of our demo)—our method is not limited to these basic actions.
>
> **(1) Diverse Interaction Scenarios in Benchmark:**
> Our benchmark includes a broad range of interaction scenarios involving people and animals. Typical examples include:
> - **Passing objects to another person (i.e., passing basketball/football)**
> - **Dancing with someone**
> - **Petting or playing with animals (i.e., riding a horse and spinning in circles with a dog)**
> - **Feeding animals**
>
> On these diverse and realistic human-human and human-animal interaction cases, our method achieves a significant performance advantage compared to existing baselines. This demonstrates the strong generalization capability and robustness of our approach. We will highlight these benchmark results and add more qualitative and quantitative examples in the revised version.
>
> **(2) Complex and Challenging Environmental Interactions:**
> Beyond interactions with people and animals, our method also handles a wide variety of complex actions involving the environment. For example:
> - **Figure 7:** Challenging actions such as diving
> - **Figure 5:** Climbing over a wall and kicking a ball
> - **Figure 8:** Precise actions like picking up milk from a cabinet
>
> Our demo video provides further examples, including:
> - **Driving by operating the steering wheel** (see 0:32 and 0:38)
> - **Riding a bicycle, swimming, climbing over a wall (1:03), and swinging a sword (0:47)**
>
> **(3) Ongoing Enrichment:**
> We are committed to further expanding the diversity and complexity of our demonstrations. In the revised version, we will provide additional visualizations featuring even more complex and varied interactions, including richer cases involving people and animals, to better illustrate the generalization and flexibility of our approach.
>
> ---
>
> **Q4.Noticeable artifacts, particularly during fast or complex motion.**
>
> **A4:** Thank you for your valuable feedback. The artifacts are mainly due to the limitations of the current adopted base model and replacing it with a stronger backbone can alleviate such issues. Specifically, when using the **Wan 2.1 14B** as our backbone, we can achieve better performance:
>
> | Model | DINO-Score (↑) | CLIP-Score (↑) | MPJPE (↓) | MRRPE (↓) | PSNR (↑) | FVD (↓) | LPIPS (↓) |
> |:-----------:|:--------------:|:--------------:|:---------:|:---------:|:--------:|:--------:|:---------:|
> | 1.3B        |    67.8        |    88.2        | 127.16    | 151.62    |  52.6    | 226.12   | 0.0663    |
> | 14B         |    69.1        |    90.7        | 109.34    | 124.20    |  56.0    | 182.55   | 0.0515    |
>
> Thus if more powerful backbone architectures become available, we will replace our current backbone in future work. This upgrade could further reduce potential artifacts and improve the fidelity of the generated results.
>
> ---
>
> **Q5:Potential benefits of incorporating more images or scene information as input to the model.**
>
> **A5:** We would like to clarify that our current framework is intentionally designed to operate with only a single egocentric first frame and related human motion provided by the user. This design choice was made to simplify the pipeline and ensure a lightweight, user-friendly experience, but it does limit our ability to directly utilize additional images or scene inputs in the present version.
>
> We agree that introducing richer scene information—such as additional frames, multi-view images, or more detailed environmental cues—could potentially improve the temporal consistency and visual realism of the generated video. Exploring how to effectively integrate such supplementary scene inputs is a promising direction for future work, and we appreciate the reviewer’s suggestion. In future versions, we plan to extend our framework to support and benefit from richer scene context, and we expect this will further enhance the fidelity and robustness of egocentric video generation.

---

### Comment · Area_Chair_qztD · 2025-08-05
**Rebuttal posted - Please engage in discussion**

Hi Reviewers,

The author has posted their rebuttal. What are your thoughts on their response? Please engage in the discussion with the author as soon as possible. The deadline for discussion is August 8th.

Thanks,

Your AC

---

### Note · Authors · 2025-08-16

Dear Area Chair and Reviewers,

We sincerely thank you for the thoughtful reviews and constructive discussions. We appreciate the recognition of novelty, a well-designed data pipeline, strong results, and clear writing for PlayerOne’s egocentric world simulation with part-disentangled motion control and scene–frame reconstruction.

What we clarified and strengthened in rebuttal & discussion:

- **Long-horizon generation & consistency**. We clarified that, after Causvid distillation, PlayerOne generates videos autoregressively for >30s (not by clip concatenation) and we added a 24s benchmark, showing comparable metrics and strong scene consistency over long sequences; these additions directly address concerns about temporal horizon and point-map consistency.

- **Latency for interactive use**. We reported the end-to-end per-frame latency of ~119 ms (≈8.4 FPS) from motion input to video frame, including all processing, to better gauge deployability.

- **Failure modes & limitations**. We provided side-by-side evaluation on game-like vs. real-world domains and discussed artifacts under fast/complex motions, the effect of stronger backbones, and a structural limitation—no explicit physical interaction modeling (e.g., collision/contact). We will expand the Limitations section and add visualizations accordingly.

- **Distillation details & reproducibility**. We added concrete settings for the Causvid distillation stage (e.g., 8,000 ODE pairs; 30k + 30k iterations with AdamW at 5e-6 / 2e-6; DMD loss; guidance scale 3.5; two-time-scale update rule; 5 days on 32×A800), and noted that quality is largely maintained; full code and scripts will be released.

- **Clarifications to design & writing**. We elaborated how PMI enables compositional, part-level control, how scene latents are aligned and used autoregressively to keep geometry consistent, and why our task setting is distinct and practically valuable; we will further polish these explanations in the revision.

We are grateful that reviewers found our responses helpful. We will integrate the added analyses, longer-horizon results, latency reporting, distillation details, and expanded limitations into the revised manuscript.

Once again, thanks for the rigorous process and for helping us improve PlayerOne. We believe the clarified design and new evidence strengthen the paper’s claims on precise motion control, world consistency, and readiness for interactive scenarios. (:D)

---

### Decision · Program_Chairs · 2025-09-17

**Decision:**

Accept (oral)

**Comment:**

This paper presents PlayerOne, a novel egocentric world simulator that constructs dynamic environments from a single image and generates videos precisely aligned with real-world human motion. The work addresses a significant and novel problem with a sound and sensible methodology, and is considered highly valuable to the community.

All reviewers and the Area Chair unanimously recommended acceptance. We strongly encourage the authors to integrate the rebuttal discussions and reviewer suggestions into the final version to maximize the paper's impact.